# KungfuBot: Physics-Based Humanoid Whole-Body Control for Learning Highly-Dynamic Skills

**Weiji Xie**[* 1,2]    **Jinrui Han**[* 1,2]    **Jiakun Zheng**[* 1,3]    **Huanyu Li**[1,4]    **Xinzhe Liu**[1,5]
**Jiyuan Shi**[1]    **Weinan Zhang**[2]    **Chenjia Bai**[† 1]    **Xuelong Li**[† 1]

[1]Institute of Artificial Intelligence (TeleAI), China Telecom
[2]Shanghai Jiao Tong University    [3]East China University of Science and Technology
[4]Harbin Institute of Technology    [5]ShanghaiTech University

## Abstract

Humanoid robots are promising to acquire various skills by imitating human behaviors. However, existing algorithms are only capable of tracking smooth, low-speed human motions, even with delicate reward and curriculum design. This paper presents a physics-based humanoid control framework, aiming to master highly-dynamic human behaviors such as Kungfu and dancing through multi-steps motion processing and adaptive motion tracking. For motion processing, we design a pipeline to extract, filter out, correct, and retarget motions, while ensuring compliance with physical constraints to the maximum extent. For motion imitation, we formulate a bi-level optimization problem to dynamically adjust the tracking accuracy tolerance based on the current tracking error, creating an adaptive curriculum mechanism. We further construct an asymmetric actor-critic framework for policy training. In experiments, we train whole-body control policies to imitate a set of highly-dynamic motions. Our method achieves significantly lower tracking errors than existing approaches and is successfully deployed on the Unitree G1 robot, demonstrating stable and expressive behaviors. The project page is `https://kungfu-bot.github.io`.

## 1 Introduction

Humanoid robots, with their human-like morphology, have the potential to mimic various human behaviors in performing different tasks [1]. The ongoing advancement of motion capture (MoCap) systems and motion generation methods has led to the creation of extensive motion datasets [2, 3], which encompass a multitude of human activities annotated with textual descriptions [4]. Consequently, it becomes promising for humanoid robots to learn whole-body control to imitate human behaviors. However, controlling high-dimensional robot actions to achieve ideal human-like performance presents a substantial challenge. One major difficulty arises from the fact that motion sequences captured from humans may not comply with the physical constraints of humanoid robots, including joint limits, dynamics, and kinematics [5, 6]. Hence, directly training policies through Reinforcement Learning (RL) to maximize rewards (e.g., the negative tracking error) often fails to yield desirable policies, as it may not exist within the solution space.

Recently, several RL-based whole-body control frameworks have been proposed to track motions [7, 8], which often take a reference kinematic motion as input and output the control actions for a humanoid robot to imitate it. To address physical feasibility issues, H2O and OmniH2O [9, 10] remove the infeasible motions using a trained privileged imitation policy, producing a clean motion

---

[*]Equal contributions.
[†]Correspondence to: Chenjia Bai (baicj@chinatelecom.cn)

39th Conference on Neural Information Processing Systems (NeurIPS 2025).

dataset. ExBody [7] constructs a feasible motion dataset by filtering via language labels, such as 'wave' and 'walk'. Exbody2 [5] trains an initial policy on all motions and uses the tracking error to measure the difficulty of each motion. However, it would be costly to train the initial policy and find an optimal dataset. There is also a lack of suitable tolerance mechanisms for difficult-to-track motions in the training process. As a result, previous methods are only capable of tracking low-speed and smooth motions. Recently, ASAP [6] introduces a multi-stage mechanism and learned a residual policy to compensate for the sim-to-real gap, reducing the difficulties in tracking agile motions. Unlike ASAP, we focus on improving motion feasibility and agility entirely in simulation.

In this paper, we propose *Physics-Based Humanoid motion Control (PBHC)*, which utilizes a two-stage framework to tackle the challenges associated with agile and highly-dynamic motions. (i) In the motion processing stage, we first extract motions from videos and establish physics-based metrics to filter out human motions by estimating physical quantities within the human model, thereby eliminating motions beyond the physical limits. Then, we compute contact masks of motions followed by motion correction, and finally retarget processed motions to the robot using differential inverse kinematics. (ii) In the motion imitation stage, we propose an adaptive motion tracking mechanism that adjusts the tracking reward via a tracking factor. Perfectly tracking hard motions is impractical due to imperfect reference motions and the need of smooth control, so we adapt the tracking factor to different motions based on the tracking error. We then formulate a Bi-Level Optimization (BLO) [11] to derive the optimal factor and design an adaptive update rule that estimates the tracking error online to dynamically refine the factor during training.

Building on the two-stage framework, we design an asymmetric actor-critic architecture for policy optimization. The critic adopts a reward vectorization technique and leverages privileged information to improve value estimation, while the actor relies solely on local observations. In experiments, PBHC enables whole-body control policies to track highly-dynamic motions with lower tracking errors than existing methods. We further demonstrate successful real-world deployment on the Unitree G1 robot, achieving stable and expressive behaviors, including complex motions like Kungfu and dancing.

## 2  Preliminaries

**Problem Formulation.** We adopt the Unitree G1 robot [12] in our work, which has 23 degrees of freedom (DoFs) to control, excluding the 3 DoFs in each wrist of the hand. We formulate the motion imitation problem as a goal-conditional RL problem with Markov Decision Process $\mathcal{M} = (\mathcal{S}, \mathcal{A}, \mathcal{S}^{\text{ref}}, \gamma, r, P)$, where $\mathcal{S}$ and $\mathcal{S}^{\text{ref}}$ are the state spaces of the humanoid robot and reference motion, respectively, $\mathcal{A}$ is the robot's action space, $r$ is a mixed reward function consisting motion-tracking and regularization rewards, and $P$ is the transition function depending on the robot morphology and physical constraints. At each time step $t$, the policy $\pi$ observes the proprioceptive state $s_t^{\text{prop}}$ of the robot and generates action $a_t$, with the aim of obtaining the next-state $s_{t+1}$ that follows the corresponding reference state $s_{t+1}^{\text{ref}}$ in the reference trajectory $[s_0^{\text{ref}}, \dots, s_{N-1}^{\text{ref}}]$. The action $a_t \in \mathbb{R}^{23}$ is the target joint position for a PD controller to compute the motor torques. We adopt an off-the-shelf RL algorithm, PPO [13], for policy optimization with an actor-critic architecture.

**Reference Motion Processing.** For human motion processing, the Skinned Multi-Person Linear (SMPL) model [14] offers a general representation of human motions, using three key parameters: $\boldsymbol{\beta} \in \mathbb{R}^{10}$ for body shapes, $\boldsymbol{\theta} \in \mathbb{R}^{24 \times 3}$ for joint rotations in axis-angle representation, and $\boldsymbol{\psi} \in \mathbb{R}^3$ for global translation. These parameters can be mapped to a 3D mesh consisting of 6,890 vertices via a differentiable skinning function $M(\cdot)$, which formally expressed as $\mathcal{V} = M(\boldsymbol{\beta}, \boldsymbol{\theta}, \boldsymbol{\psi}) \in \mathbb{R}^{6890 \times 3}$. We employ a human motion recovery model to estimate SMPL parameters $(\boldsymbol{\beta}, \boldsymbol{\theta}, \boldsymbol{\psi})$ from videos, followed by additional motion processing. The resulting SMPL-format motions are then retargeted to G1 through an Inverse Kinematics (IK) method, yielding the reference motions for tracking purposes.

## 3  Methods

An overview of PBHC is illustrated in Fig. 1. First, raw human videos are processed by a Human Motion Recovery (HMR) model to produce SMPL-format motion sequences. These sequences are filtered via physics-based metrics and corrected using contact masks. The refined motions are then retargeted to the G1 robot. Finally, each resulting trajectory serves as reference motion for training a separate RL policy, which is then deployed on the real G1 robot. In the following, we detail the motion processing pipeline (§3.1), adaptive motion tracking module (§3.2) and RL framework (§3.3).

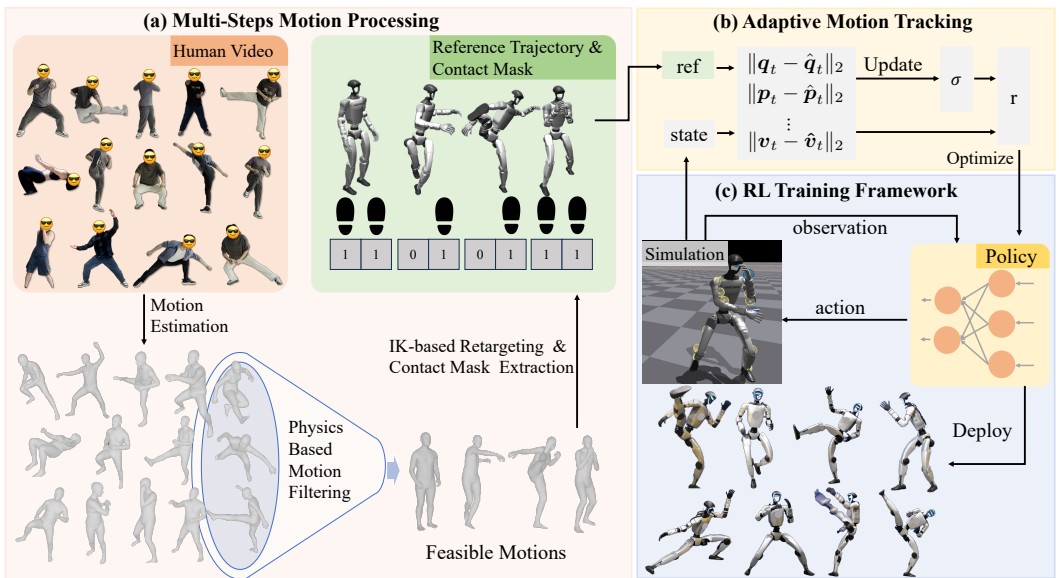

Figure 1: An overview of PBHC that includes three core components: (a) motion extraction from videos and multi-steps motion processing, (b) adaptive motion tracking based on the optimal tracking factor, (c) the RL training framework and sim-to-real deployment.

## 3.1 Motion Processing Pipeline

We propose a motion processing pipeline to extract motion from videos for humanoid motion tracking, comprising four steps: (i) SMPL-format motion estimation from monocular videos, (ii) physics-based motion filtering, (iii) contact-aware motion correction, and (iv) motion retargeting. This pipeline ensures that physically plausible motions can be transferred from videos to humanoid robots.

**Motion Estimation from Videos.** We employ GVHMR [15] to estimate SMPL-format motions from monocular videos. GVHMR introduces a gravity-view coordinate system that naturally aligns motions with gravity, eliminating body tilt issues caused by reconstruction solely relying on the camera coordinate system. Furthermore, it mitigates foot sliding artifacts by predicting foot stationary probabilities, thereby enhancing motion quality.

**Physics-based Motion Filtering.** Due to reconstruction inaccuracies and out-of-distribution issues in HMR models, motions extracted from videos may violate physical and biomechanical constraints. Thus, we try to filter out these motions via physics-based principles. Previous work [16] suggests that proximity between the center of mass (CoM) and center of pressure (CoP) indicates greater stability, and proposes a method to estimate CoM and CoP coordinates from SMPL data. Building on this, we calculate the projected distance of CoM and CoP on the ground for each frame and apply a threshold to assess stability. Specifically, let $\bar{\boldsymbol{p}}_t^{\mathrm{CoM}} = (p_{t,x}^{\mathrm{CoM}}, p_{t,y}^{\mathrm{CoM}})$ and $\bar{\boldsymbol{p}}_t^{\mathrm{CoP}} = (p_{t,x}^{\mathrm{CoP}}, p_{t,y}^{\mathrm{CoP}})$ denote the projected coordinates of CoM and CoP on the ground at frame $t$ respectively, and $\Delta d_t$ represents the distance between these projections. We define the stability criterion of a frame as

$$\Delta d_t = \|\bar{\boldsymbol{p}}_t^{\mathrm{CoM}} - \bar{\boldsymbol{p}}_t^{\mathrm{CoP}}\|_2 < \epsilon_{\mathrm{stab}}, \tag{1}$$

where $\epsilon_{\mathrm{stab}}$ represents the stability threshold. Then, given an $N$-frame motion sequence, let $\mathcal{B} = [t_0, t_1, \ldots, t_K]$ be the increasingly sorted list of frame indices that satisfy Eq. (1), where $t_k \in [1, N]$. The motion sequence is considered stable if it satisfies two conditions: (i) Boundary-frame stability: $1 \in \mathcal{B}$ and $N \in \mathcal{B}$. (ii) Maximum instability gap: the maximum length of consecutive unstable frames must be less than threshold $\epsilon_{\mathrm{N}}$, i.e., $\max_k t_{k+1} - t_k < \epsilon_{\mathrm{N}}$. Based on this criterion, motions that are clearly unable to maintain dynamic stability can be excluded from the original dataset.

**Motion Correction based on Contact Mask.** To better capture foot-ground contact in motion data, we estimate contact masks by analyzing ankle displacement across consecutive frames, based on the zero-velocity assumption [17, 18]. Let $\boldsymbol{p}_t^{\text{l-ankle}} \in \mathbb{R}^3$ denote the position of the left ankle joint at time $t$, and $c_t^{\text{left}} \in \{0, 1\}$ the corresponding contact mask. The contact mask is estimated as

$$c_t^{\text{left}} = \mathbb{I}[\|\boldsymbol{p}_{t+1}^{\text{l-ankle}} - \boldsymbol{p}_t^{\text{l-ankle}}\|_2^2 < \epsilon_{\text{vel}}] \cdot \mathbb{I}[p_{t,z}^{\text{l-ankle}} < \epsilon_{\text{height}}], \tag{2}$$

where $\epsilon_{\text{vel}}$ and $\epsilon_{\text{height}}$ are empirically chosen thresholds. Similarly for the right foot.

To address minor floating artifacts not eliminated by threshold-based filtering, we apply a correction step based on the estimated contact mask. Specifically, if either foot is in contact at frame $t$, a vertical offset is applied to the global translation. Let $\boldsymbol{\psi}_t$ denotes the global translation of the pose at time $t$, then the corrected vertical position is:

$$\psi_{t,z}^{\text{corr}} = \psi_{t,z} - \Delta h_t, \tag{3}$$

where $\Delta h_t = \min_{v \in \mathcal{V}_t} p_{t,z}^v$ is the lowest $z$-coordinate among the SMPL mesh vertices $\mathcal{V}_t$ at frame $t$. While the correction alleviates floating artifacts, it may cause frame-to-frame jitter. We address this by applying Exponential Moving Average (EMA) to smooth the motion.

**Motion Retargeting.** We adopt an inverse kinematics (IK)-based method [19] to retarget processed SMPL-format motions to the G1 robot. This approach formulates a differentiable optimization problem that ensures end-effector trajectory alignment while respecting joint limits.

To enhance motion diversity, we incorporate additional data from open-source datasets, AMASS [4] and LAFAN [20]. These motions are partially processed through our pipeline, including contact mask estimation, motion correction, and retargeting.

### 3.2 Adaptive Motion Tracking

#### 3.2.1 Exponential Form Tracking Reward

The reward function in PBHC, detailed in Appendix C.2, comprises two components: task-specific rewards, which enforce accurate tracking of reference motions, and regularization rewards, which promote overall stability and smoothness.

The task-specific rewards include terms for aligning joint states, rigid body state, and foot contact mask. These rewards, except the foot contact tracking term, follow the exponential form as:

$$r(x) = \exp(-x/\sigma), \tag{4}$$

where $x$ represents the tracking error, typically measured as the mean squared error (MSE) of quantities such as joint angles, while $\sigma$ controls the tolerance of the error, referred to as the *tracking factor*. This exponential form is preferred over the negative error form because it is bounded, helps stabilize the training process, and provides a more intuitive approach for reward weighting.

Intuitively, when $\sigma$ is much larger than the typical range of $x$, the reward remains close to 1 and becomes insensitive to changes in $x$, while an overly small $\sigma$ causes the reward to approach 0 and also reduces its sensitivity, highlighting the importance of choosing $\sigma$ appropriately to enhance responsiveness and hence tracking precision. This intuition is illustrated in Fig. 2.

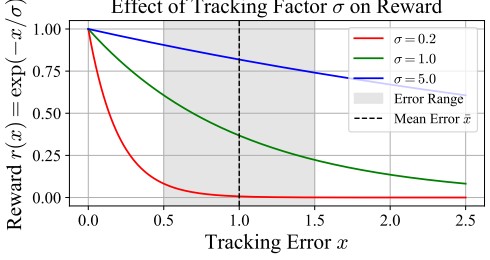

Figure 2: Illustration of the effect of tracking factor $\sigma$ on the reward value.

#### 3.2.2 Optimal Tracking Factor

To determine the choice of the optimal tracking factor, we introduce a simplified model of motion tracking and formulate it as a bi-level optimization problem. The intuition behind this formulation is that **the tracking factor $\sigma$ should be chosen to minimize the accumulated tracking error of the converged policy over the reference trajectory**. In manual tuning scenarios, this is typically achieved through an iterative process where an engineer selects a value for $\sigma$, trains a policy, observes the results, and repeats the process until satisfactory performance is attained.

Given a policy $\pi$, there is a sequence of expected tracking error $\boldsymbol{x} \in \mathbb{R}_+^N$ for $N$ steps, where $x_i$ represents the expected tracking error at the $i$-th step of the rollout episodes. Rather than optimizing the policy directly, we treat the tracking error sequence $\boldsymbol{x}$ as decision variables. This allows us to reformulate the optimization problem of motion tracking as:

$$\max_{\boldsymbol{x} \in \mathbb{R}_+^N} J^{\text{in}}(\boldsymbol{x}, \sigma) + R(\boldsymbol{x}), \tag{5}$$

where the *internal* objective $J^{\text{in}}(\boldsymbol{x}, \sigma) = \sum_{i=1}^{N} \exp(-x_i/\sigma)$ is the simplified accumulated reward induced by the tracking reward in Eq. (4), and we introduce $R(\boldsymbol{x})$ to capture all additional effects beyond $J^{\text{in}}$, including environment dynamics and other policy objectives such as extra rewards. The solution $\boldsymbol{x}^*$ to Eq. (5) corresponds to the error sequence induced by the optimal policy $\pi^*$. Subsequently, the optimization objective of $\sigma$ is to maximize the obtained accumulated negative tracking error $J^{\text{ex}}(\boldsymbol{x}^*) = \sum_{i=1}^{N} -x_i^*$, the *external* objective, formalized as the following bi-level optimization problem:

$$\max_{\sigma \in \mathbb{R}_+} \quad J^{\text{ex}}(\boldsymbol{x}^*), \qquad \text{s.t.} \quad \boldsymbol{x}^* \in \arg \max_{\boldsymbol{x} \in \mathbb{R}_+^N} J^{\text{in}}(\boldsymbol{x}, \sigma) + R(\boldsymbol{x}). \tag{6}$$

This simplified modeling provides an intuitive connection to the RL training process.

- The lower-level optimization represents the standard RL procedure, where a policy is trained to maximize tracking reward and other reward terms, given a specific $\sigma$.
- The upper-level optimization, outside the RL loop, selects $\sigma$ to minimize the total tracking error of the final converged policy. This outer optimization is not reward maximization but a performance-driven objective based on absolute external metrics.

Under additional technical assumptions, we can solve Eq. (6) and derive that the optimal tracking factor is the average of the optimal tracking error, as detailed in Appendix A.

$$\sigma^* = \left( \sum_{i=1}^{N} x_i^* \right) / N. \tag{7}$$

### 3.2.3 Adaptive Mechanism

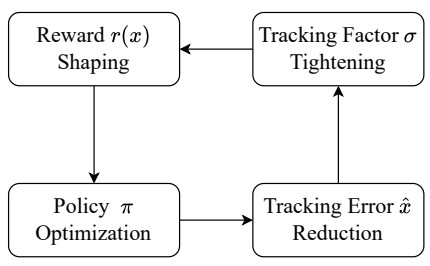

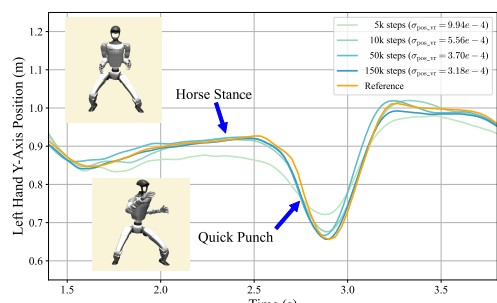

Figure 3: Closed-loop adjustment of tracking factor in the proposed adaptive mechanism.

Figure 4: Example of the right hand $y$-position for 'Horse-stance punch'. The adaptive $\sigma$ can progressively improve the tracking precision. $\sigma_{\text{pos\_vr}}$ is used for tracking the head and hands.

While Eq. (7) provides a theoretical guidance for determining the tracking factor, the coupling between $\sigma^*$ and $\boldsymbol{x}^*$ creates a circular dependency that prevents direct computation. Additionally, due to the varying quality and complexity of reference motion data, selecting a single, fixed value for the tracking factor that works for all motion scenarios is impractical. To resolve this, we design an adaptive mechanism that dynamically adjusts $\sigma$ during training through a feedback loop between error estimation and tracking factor adaptation.

In this mechanism, we maintain an Exponential Moving Average (EMA) $\hat{x}$ of the instantaneous tracking error over environment steps. This EMA serves as an online estimate of the expected tracking error under the current policy, and during training this value should approach the average optimal tracking error $\left( \sum_{i=1}^{N} x_i^* \right) / N$ under the current factor $\sigma$. At each step, PBHC updates $\sigma$ to the current value of $\hat{x}$, creating a feedback loop where reductions in tracking error lead to tightening of $\sigma$. This closed-loop process drives further policy refinement, and as the tracking error decreases, the system converges to an optimal value of $\sigma$ that asymptotically solves Eq. (9), as illustrated in Fig. 3.

To ensure stability during training, we constrain $\sigma$ to be non-increasing and initialize it with a relatively large value, $\sigma^{\text{init}}$. The update rule is given by Eq. (8). As shown in Fig. 4, this adaptive mechanism allows the policy to progressively improve its tracking precision during training.

$$\sigma \leftarrow \min(\sigma, \hat{x}). \tag{8}$$

## 3.3 RL Training Framework

**Asymmetric Actor-Critic.** Following previous works [6, 21], the time phase variable $\phi_t \in [0, 1]$ is introduced to represent the current progress of the reference motion linearly, where $\phi_t = 0$ denotes the start of a motion and $\phi_t = 1$ denotes the end. The observation of the actor $s_t^{\text{actor}}$ includes the robot's proprioception $s_t^{\text{prop}}$ and the time phase variable $\phi_t$. The proprioception $s_t^{\text{prop}} = [\boldsymbol{q}_{t-4:t}, \dot{\boldsymbol{q}}_{t-4:t}, \boldsymbol{\omega}_{t-4:t}^{\text{root}}, \boldsymbol{g}_{t-4:t}^{\text{proj}}, \boldsymbol{a}_{t-5:t-1}]$ includes 5-step history of joint position $\boldsymbol{q}_t \in \mathbb{R}^{23}$, joint velocity $\dot{\boldsymbol{q}}_t \in \mathbb{R}^{23}$, root angular velocity $\boldsymbol{\omega}_t^{\text{root}} \in \mathbb{R}^3$, root projected gravity $\boldsymbol{g}_t^{\text{proj}} \in \mathbb{R}^3$ and last-step action $\boldsymbol{a}_{t-1} \in \mathbb{R}^{23}$. The critic receives an augmented observation $s_t^{\text{crtic}}$, including $s_t^{\text{prop}}$, time phase, reference motion positions, root linear velocity, and a set of randomized physical parameters.

**Reward Vectorization.** To facilitate the learning of value function with multiple rewards, we vectorize rewards and value functions as: $\boldsymbol{r} = [r_1, \ldots, r_n]$ and $\boldsymbol{V}(s) = [V_1(\boldsymbol{s}), \ldots, V_n(\boldsymbol{s})]$ following Xie et al. [22]. Rather than aggregating all rewards into a single scalar, each reward component $r_i$ is assigned to a value function $V_i(\boldsymbol{s})$ that independently estimates returns, implemented by a critic network with multiple output heads. All value functions are aggregated to compute the action advantage. This design enables precise value estimation and promotes stable policy optimization.

**Reference State Initialization.** We use Reference State Initialization (RSI) [21], which initializes the robot's state from reference motion states at randomly sampled time phases. This facilitates parallel learning of different motion phases, significantly improving training efficiency.

**Sim-to-Real Transfer.** To bridge the sim-to-real gap, we adopt domain randomization by varying the physical parameters of the simulated environment and humanoids. The trained policies are validated through sim-to-sim testing before being directly deployed to real robots, achieving zero-shot sim-to-real transfer without any fine-tuning. Details are in Appendix C.3.

# 4 Related Works

**Humanoid Motion Imitation.** Robot motion imitation aims to learn lifelike and natural behaviors from human motions [21, 23]. Although there exist several motion datasets that contain diverse motions [24, 25, 4], humanoid robots cannot directly learn the diverse behaviors due to the significantly different physical structures between humans and humanoid robots [6, 26]. Meanwhile, most datasets lack physical information, such as foot contact annotations that would be important for robot policy learning [27, 28]. As a result, we adopt physics-based motion processing for motion filtering and contact annotation. After obtaining the reference motion, the humanoid robot learns a whole-body control policy to interact with the simulator [29, 30], with the aim of obtaining a state trajectory close to the reference [31, 32]. However, learning such a policy is quite challenging, as the robot requires precise control of high-dimensional DoFs to achieve stable and realistic movement [7, 8]. Recent advances adopt physics-based motion filtering and RL to learn whole-body control policies [5, 10], and perform real-world adaptation via sim-to-real transfer [33]. However, because of the lack of tolerance mechanisms for hard motions, these methods are only capable of tracking relatively simple motions. Other works also combine teleoperation [34, 35] and independent control of upper and lower bodies [36], while they may sacrifice the expressiveness of motions. In contrast, we propose an adaptive mechanism to dynamically adapt the tracking rewards for agile motions.

**Humanoid Whole-Body Control.** Traditional methods for humanoid robots usually learn independent control policies for locomotion and manipulation. For the lower-body, RL-based controller have been widely adopted to learn locomotion policies for complex tasks such as complex-terrain walking [37, 38], gait control [39], standing up [40, 41], jumping [42], and even parkour [43, 44]. However, each locomotion task requires delicate reward designs, and human-like behaviors are difficult to obtain [45, 46]. In contrast, we adopt human motion as references, which is straightforward for robots to obtain human-like behaviors. For the upper-body, various methods propose different architectures to learn manipulation tasks, such as diffusion policy [47, 48], visual-language-action model [49, 50, 51], dual-system architecture [52, 53], and world models [54, 55]. However, these methods may overlook the coordination of the two limbs. Recently, several whole-body control methods have been proposed, with the aim of enhancing the robustness of entire systems in locomotion [22, 39, 34] or performing loco-manipulation tasks [56]. Differently, the upper and lower bodies of our method have the same objective to track the reference motion, while the lower body still requires maintaining stability and preventing falling in motion imitation. Other methods collect whole-body control datasets to learn a

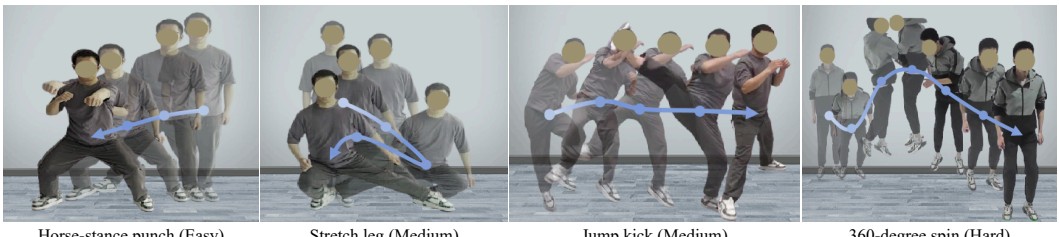

| Horse-stance punch (Easy) | Stretch leg (Medium) | Jump kick (Medium) | 360-degree spin (Hard) |

Figure 5: Example motions in our constructed dataset. Darker opacity indicates later timestamps.

humanoid foundation model [56, 57], while requiring a large number of trajectories. In contrast, we only require a small number of reference motions to learn diverse behaviors.

# 5 Experiments

In this section, we present experiments to evaluate the effectiveness of PBHC. Our experiments aim to answer the following key research questions:

- **Q1.** Can our physics-based motion filtering effectively filter out untrackable motions?
- **Q2.** Does PBHC achieve superior tracking performance compared to prior methods in simulation?
- **Q3.** Does the adaptive motion tracking mechanism improve tracking precision?
- **Q4.** How well does PBHC perform in real-world deployment?

## 5.1 Experiment Setup

**Evaluation Method.** We assess the policy's tracking performance using a highly-dynamic motion dataset constructed through our proposed motion processing pipeline, detailed in Appendix B. Examples are shown in Fig. 5. We categorize motions into three difficulty levels: easy, medium, and hard, based on their agility requirements. For each setting, policies are trained in IsaacGym [29] with three random seeds and are evaluated over 1,000 rollout episodes.

**Metrics.** The tracking performance of polices is quantified through the following metrics: Global Mean Per Body Position Error ($E_{\text{g-mpbpe}}$, mm), root-relative Mean Per Body Position Error ($E_{\text{mpbpe}}$, mm), Mean Per Joint Position Error ($E_{\text{mpjpe}}$, $10^{-3}$ rad), Mean Per Joint Velocity Error ($E_{\text{mpjve}}$, $10^{-3}$ rad/frame), Mean Per Body Velocity Error ($E_{\text{mpbve}}$, mm/frame), and Mean Per Body Acceleration Error ($E_{\text{mpbae}}$, mm/frame$^2$). The definition of metrics is given in Appendix D.2.

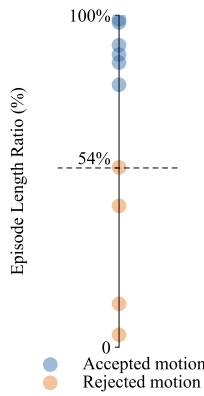

Figure 6: The distribution of ELR of accepted and rejected motions.

## 5.2 Motion Filtering

To address **Q1**, we apply our physics-based motion filtering method (see §3.1) to 10 motion sequences. Among them, 4 sequences are rejected based on the filtering criteria, while the remaining 6 are accepted. To evaluate the effectiveness of the filtering, we train a separate policy for each motion and compute the Episode Length Ratio (ELR), defined as the ratio of average episode length to reference motion length.

As shown in Fig. 6, accepted motions consistently achieve high ELRs, demonstrating motions that satisfy the physics-based metric can lead to better performance in motion tracking. In contrast, rejected motions achieve a maximum ELR of only 54%, suggesting frequent violations of termination conditions. These results demonstrate that our filtering method effectively excludes inherently untrackable motions, thereby improving efficiency by focusing on viable candidates.

Table 1: Main results comparing different methods across difficulty levels. PBHC consistently outperforms deployable baselines and approaches oracle-level performance. Results are reported as mean $\pm$ one standard deviation. Bold indicates methods within one standard deviation of the best result, excluding Oracle baselines. Asterisks (*) denote significant improvements ($p < 0.05$) of our method over baselines per two-sided permutation tests.

| Method | $E_{\text{g-mpbpe}} \downarrow$ | $E_{\text{mpbpe}} \downarrow$ | $E_{\text{mpjpe}} \downarrow$ | $E_{\text{mpbve}} \downarrow$ | $E_{\text{mpbae}} \downarrow$ | $E_{\text{mpjve}} \downarrow$ |
|---|---|---|---|---|---|---|
| **Easy** | | | | | | |
| OmniH2O | $233.54_{\pm4.013}{}^{*}$ | $103.67_{\pm1.912}{}^{*}$ | $1805.10_{\pm12.33}{}^{*}$ | $8.54_{\pm0.125}{}^{*}$ | $8.46_{\pm0.081}{}^{*}$ | $224.70_{\pm2.043}$ |
| ExBody2 | $588.22_{\pm11.43}{}^{*}$ | $332.50_{\pm3.584}{}^{*}$ | $4014.40_{\pm21.50}{}^{*}$ | $14.29_{\pm0.172}{}^{*}$ | $9.80_{\pm0.157}{}^{*}$ | $206.01_{\pm1.346}{}^{*}$ |
| Ours | $\mathbf{53.25}_{\pm17.60}$ | $\mathbf{28.16}_{\pm6.127}$ | $\mathbf{725.62}_{\pm16.20}$ | $\mathbf{4.41}_{\pm0.312}$ | $\mathbf{4.65}_{\pm0.140}$ | $\mathbf{81.28}_{\pm2.052}$ |
| MaskedMimic (Oracle) | $41.79_{\pm1.715}$ | $21.86_{\pm2.030}$ | $739.96_{\pm19.94}{}^{*}$ | $5.20_{\pm0.245}$ | $7.40_{\pm0.333}{}^{*}$ | $132.01_{\pm8.941}{}^{*}$ |
| Ours (Oracle) | $45.02_{\pm6.760}$ | $22.95_{\pm15.22}$ | $710.30_{\pm16.66}$ | $4.63_{\pm1.580}$ | $4.89_{\pm0.960}$ | $73.44_{\pm12.42}$ |
| **Medium** | | | | | | |
| OmniH2O | $433.64_{\pm16.22}{}^{*}$ | $151.42_{\pm7.340}{}^{*}$ | $2333.90_{\pm49.50}{}^{*}$ | $10.85_{\pm0.300}$ | $10.54_{\pm0.152}$ | $204.36_{\pm4.473}$ |
| ExBody2 | $619.84_{\pm26.16}{}^{*}$ | $261.01_{\pm1.592}{}^{*}$ | $3738.70_{\pm26.90}{}^{*}$ | $14.48_{\pm0.160}{}^{*}$ | $11.25_{\pm0.173}$ | $204.33_{\pm2.172}{}^{*}$ |
| Ours | $\mathbf{126.48}_{\pm27.01}$ | $\mathbf{48.87}_{\pm7.550}$ | $\mathbf{1043.30}_{\pm104.4}$ | $\mathbf{6.62}_{\pm0.412}$ | $\mathbf{7.19}_{\pm0.254}$ | $\mathbf{105.30}_{\pm5.941}$ |
| MaskedMimic (Oracle) | $150.92_{\pm133.4}{}^{*}$ | $61.69_{\pm46.01}{}^{*}$ | $934.25_{\pm155.0}{}^{*}$ | $8.16_{\pm1.974}$ | $10.01_{\pm0.883}{}^{*}$ | $176.84_{\pm26.14}$ |
| Ours (Oracle) | $66.85_{\pm50.29}$ | $29.56_{\pm14.53}$ | $753.69_{\pm100.2}$ | $5.34_{\pm0.425}$ | $6.58_{\pm0.291}$ | $82.73_{\pm3.108}$ |
| **Hard** | | | | | | |
| OmniH2O | $446.17_{\pm12.84}$ | $\mathbf{147.88}_{\pm4.142}$ | $1939.50_{\pm23.90}$ | $14.98_{\pm0.643}$ | $\mathbf{14.40}_{\pm0.580}$ | $190.13_{\pm8.211}$ |
| ExBody2 | $689.68_{\pm11.80}$ | $246.40_{\pm1.252}{}^{*}$ | $4037.40_{\pm16.70}{}^{*}$ | $19.90_{\pm0.210}$ | $16.72_{\pm0.160}$ | $254.76_{\pm3.409}{}^{*}$ |
| Ours | $\mathbf{290.36}_{\pm139.1}$ | $\mathbf{124.61}_{\pm53.54}$ | $\mathbf{1326.60}_{\pm378.9}$ | $\mathbf{11.93}_{\pm2.622}$ | $\mathbf{12.36}_{\pm2.401}$ | $\mathbf{135.05}_{\pm16.43}$ |
| MaskedMimic (Oracle) | $47.74_{\pm2.762}$ | $27.25_{\pm1.615}$ | $829.02_{\pm15.41}{}^{*}$ | $8.33_{\pm0.194}$ | $10.60_{\pm0.420}{}^{*}$ | $146.90_{\pm13.32}{}^{*}$ |
| Ours (Oracle) | $79.25_{\pm69.4}$ | $34.74_{\pm22.6}$ | $734.90_{\pm155.9}$ | $7.04_{\pm1.420}$ | $8.34_{\pm1.140}$ | $93.79_{\pm17.36}$ |

## 5.3 Main Result

To address **Q2**, we compare PBHC with three baseline methods: OmniH2O [10], Exbody2 [5], and MaskedMimic [23]. All baselines employ the exponential form of the reward function for tracking reference motion, as described in §3.2.1. Implementation details are provided in Appendix D.3.

As shown in Table 1, PBHC consistently outperforms the baselines OmniH2O and ExBody2 across all evaluation metrics. These improvements can be attributed to our adaptive motion tracking mechanism, which automatically adjusts tracking factors based on motion characteristics, whereas the fixed, empirically tuned parameters in the baselines fail to generalize across diverse motions. While MaskedMimic performs well on certain metrics, it is primarily designed for character animation and is not deployable for robot control, as it does not account for constraints such as partial observability and action smoothness. To enable a fair comparison, we also train an oracle version of PBHC that similarly overlooks such constraints, in the same manner as MaskedMimic.

## 5.4 Impact of Adaptive Motion Tracking Mechanism

To investigate **Q3**, we conduct an ablation study evaluating our adaptive motion tracking mechanism (§3.2) against four baseline configurations with fixed tracking factor set: *Coarse, Medium, UpperBound, LowerBound*. The tracking factors in *Coarse*, *Medium*, *UpperBound*, and *LowerBound* are roughly progressively smaller, with *LowerBound* approximately corresponding to the smallest tracking factor derived from the adaptive mechanism after training convergence, while *UpperBound* approximately corresponds to the largest. The specific configuration of baselines and the converged tracking factors of the adaptive mechanism are given in Appendix D.4.

As shown in Fig. 7, the performance of the fixed tracking factor configurations (*Coarse, Medium, LowerBound* and *UpperBound*) varies between different motion types. Specifically, while *LowerBound* and *UpperBound* achieve strong performance on certain motions, they perform suboptimally on others, indicating that no single fixed setting consistently yields optimal tracking results on all motions. In contrast, our adaptive motion tracking mechanism consistently achieves near-optimal performance across all motion types, demonstrating its effectiveness in dynamically adjusting the tracking factor to suit varying motion characteristics.

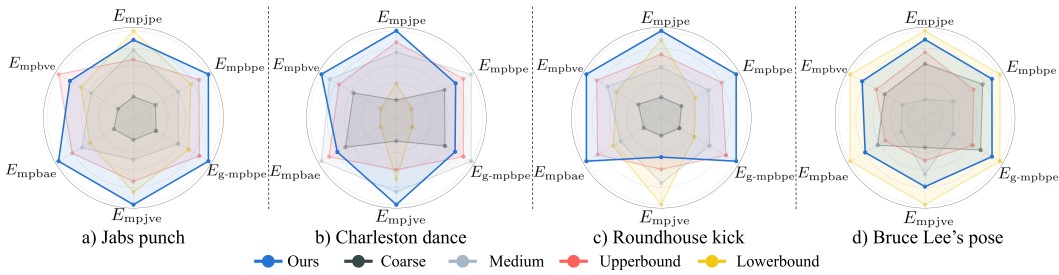

Figure 7: Ablation study comparing the adaptive motion tracking mechanism with fixed tracking factor variants. The adaptive mechanism consistently achieves near-optimal performance across all motions, whereas fixed variants exhibit varying performance depending on motions.

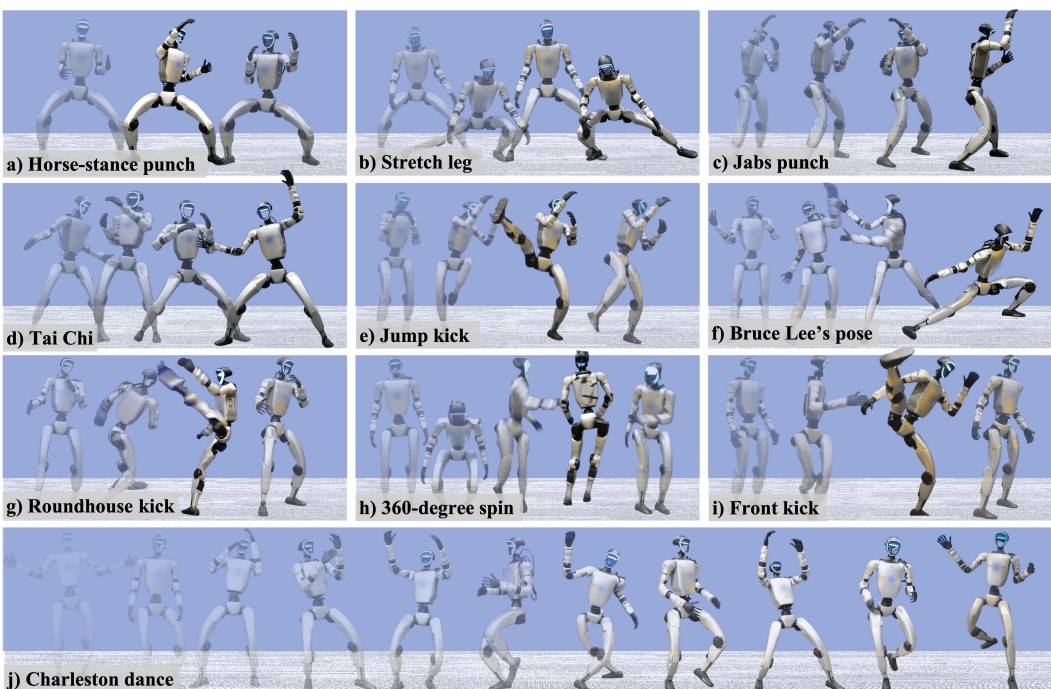

Figure 8: Our robot masters highly-dynamic skills in the real world. Time flows left to right.

## 5.5 Real-World Deployment

To investigate **Q4**, we deploy the policies in real robot. As shown in Fig. 8, 12 and the supporting videos, our robot in real world demonstrates outstanding dynamic capabilities through a diverse repertoire of advanced skills: (1) sophisticated martial arts techniques including powerful boxing combinations (jabs, hooks, and horse-stance punches) and high-degree kicking maneuvers (front kicks, jump kicks, side kicks, back kicks, and spinning roundhouse kicks); (2) acrobatic movements such as full 360-degree spins; (3) flexible motions including deep squats and stretches; (4) artistic performances ranging from dynamic dance routines to graceful Tai Chi sequences. This comprehensive skill set highlights our system's remarkable versatility, dynamic control, and real-world applicability across both athletic and artistic domains.

To quantitatively assess our policy's tracking performance, we conduct 10 trials of the Tai Chi motion and compute evaluation metrics based on the onboard sensor readings, as shown in Table 2. Notably, the metrics obtained in the real world are closely aligned with those from the sim-to-sim platform MuJoCo, demonstrating that our policy can robustly transfer from simulation to real-world deployment while maintaining high-performance control.

Table 2: Comparison of tracking performance of Tai Chi between real-world and simulation. The robot root is fixed to the origin since it's inaccessible in real-world.

| Platform | $E_{\mathrm{mpbpe}} \downarrow$ | $E_{\mathrm{mpjpe}} \downarrow$ | $E_{\mathrm{mpbve}} \downarrow$ | $E_{\mathrm{mpbae}} \downarrow$ | $E_{\mathrm{mpjve}} \downarrow$ |
|---|---|---|---|---|---|
| MuJoCo | $33.18_{\pm 2.720}$ | $1061.24_{\pm 83.27}$ | $2.96_{\pm 0.342}$ | $2.90_{\pm 0.498}$ | $67.71_{\pm 6.747}$ |
| Real | $36.64_{\pm 2.592}$ | $1130.05_{\pm 9.478}$ | $3.01_{\pm 0.126}$ | $3.12_{\pm 0.056}$ | $65.68_{\pm 1.972}$ |

## 5.6 Learning Curves

To additionally illustrate the training process and verify its stability, we present in Fig. 9 the learning curves for three representative motions—Jabs Punch, Tai Chi, and Roundhouse Kick—showing both the mean episode length and mean reward. These curves provide an intuitive view of how the policy improves over time, and it can be observed that training gradually stabilizes and converges after approximately 20k steps, demonstrating the reliability and efficiency of our approach in learning complex motion behaviors.

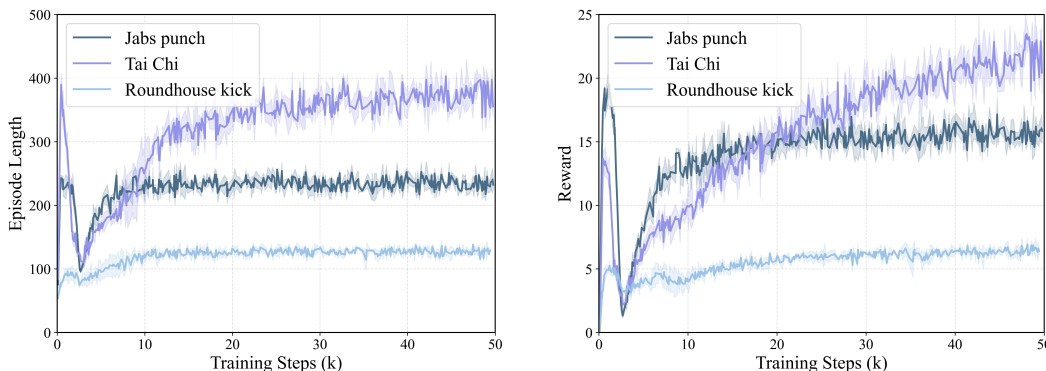

Figure 9: Mean episode length and mean reward across three motions. Both curves indicate that training gradually stabilizes after 20k steps.

## 6 Conclusion & Limitations

This paper introduces PBHC, a novel RL framework for humanoid whole-body motion control that achieves outstanding highly-dynamic behaviors and superior tracking accuracy through physics-based motion processing and adaptive motion tracking. The experiments show the motion filtering metric can efficiently filter out trajectories that are difficult to track, and the adaptive motion tracking method consistently outperforms baseline methods on tracking error. The real-world deployments demonstrate robust behaviors for athletic and artistic domains. These contributions push the boundaries of humanoid motion control, paving the way for more agile and stable real-world applications.

However, our method still has limitations. (i) It lacks environment awareness, such as terrain perception and obstacle avoidance, which restricts deployment in unstructured real-world settings. (ii) Each policy is trained to imitate a single motion, which may not be efficient for applications requiring diverse motion repertoires. We leave research on how to maintain high dynamic performance while enabling broader skill generalization for the future.

## Acknowledgments and Disclosure of Funding

This work is supported by the National Key Research and Development Program of China (Grant No.2024YFE0210900), Shanghai Municipal Science and Technology Major Project (Grant No.2021SHZDZX0102), the National Natural Science Foundation of China (Grant No.62306242 and No.62322603), the Young Elite Scientists Sponsorship Program by CAST (Grant No.2024QNRC001), and the Yangfan Project of the Shanghai (Grant No.23YF11462200).

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

# A    Derivation of Optimal Tracking Sigma

We recall the bi-level optimization problem in (6), as

$$\max_{\sigma \in \mathbb{R}_+} \quad J^{\mathrm{ex}}(\boldsymbol{x}^*) \tag{9a}$$

$$\text{s.t.} \quad \boldsymbol{x}^* \in \arg \max_{\boldsymbol{x} \in \mathbb{R}_+^N} J^{\mathrm{in}}(\boldsymbol{x}, \sigma) + R(\boldsymbol{x}) \tag{9b}$$

Assuming $R(\boldsymbol{x})$ takes a linear form $R(\boldsymbol{x}) = \boldsymbol{A}\boldsymbol{x} + \boldsymbol{b}$, $J^{\mathrm{ex}}$, and $J^{\mathrm{in}}$ are twice continuously differentiable and the lower-level problem Eq. (9b) has a unique solution $\boldsymbol{x}^*(\sigma)$. Then we take an implicit gradient approach to solve it. The gradient of $J^{\mathrm{ex}}$ w.r.t. $\sigma$ is:

$$\frac{dJ^{\mathrm{ex}}}{d\sigma} = \frac{d\boldsymbol{x}^*(\sigma)}{d\sigma}^\top \nabla_{\boldsymbol{x}} J^{\mathrm{ex}}(\boldsymbol{x}^*(\sigma)). \tag{10}$$

To obtain $\frac{d\boldsymbol{x}^*(\sigma)}{d\sigma}$, since $\boldsymbol{x}^*(\sigma)$ is a lower-level solution, it satisfies:

$$\nabla_{\boldsymbol{x}}(J^{\mathrm{in}}(\boldsymbol{x}^*(\sigma), \sigma) + R(\boldsymbol{x})) = 0. \tag{11}$$

Take the first-order derivative of Eq. (11) w.r.t. $\sigma$, then we have:

$$\frac{d}{d\sigma}(\nabla_{\boldsymbol{x}}(J^{\mathrm{in}}(\boldsymbol{x}^*(\sigma), \sigma) + R(\boldsymbol{x})) = \nabla_{\sigma, \boldsymbol{x}}^2 J^{\mathrm{in}} + \frac{d\boldsymbol{x}^*(\sigma)}{d\sigma}^\top \nabla_{\boldsymbol{x}, \boldsymbol{x}}^2 J^{\mathrm{in}} = 0, \tag{12}$$

$$\frac{d\boldsymbol{x}^*(\sigma)}{d\sigma}^\top = -\nabla_{\sigma, \boldsymbol{x}}^2 J^{\mathrm{in}}(\boldsymbol{x}^*(\sigma), \sigma) \nabla_{\boldsymbol{x}, \boldsymbol{x}}^2 J^{\mathrm{in}}(\boldsymbol{x}^*(\sigma), \sigma)^{-1}. \tag{13}$$

Substituting Eq. (13) into Eq. (10), we have

$$\frac{dJ^{\mathrm{ex}}}{d\sigma} = -\nabla_{\sigma, \boldsymbol{x}}^2 J^{\mathrm{in}}(\boldsymbol{x}^*(\sigma), \sigma) \nabla_{\boldsymbol{x}, \boldsymbol{x}}^2 J^{\mathrm{in}}(\boldsymbol{x}^*(\sigma), \sigma)^{-1} \nabla_{\boldsymbol{x}} J^{\mathrm{ex}}(\boldsymbol{x}^*(\sigma)), \tag{14}$$

where

$$J^{\mathrm{ex}}(\boldsymbol{x}) = \sum_{i=1}^N -x_i, \tag{15a}$$

$$J^{\mathrm{in}}(\boldsymbol{x}, \sigma) = \sum_{i=1}^N \exp(-x_i/\sigma). \tag{15b}$$

Compute first- and second-order gradients in Eq. (14) as

$$\nabla_{\boldsymbol{x}} J^{\mathrm{in}}(\boldsymbol{x}, \sigma) = \exp(-\boldsymbol{x}/\sigma)(-\frac{1}{\sigma}), \tag{16a}$$

$$\nabla_{\boldsymbol{x}} J^{\mathrm{ex}}(\boldsymbol{x}) = \mathbf{1}, \tag{16b}$$

$$\nabla_{\sigma, \boldsymbol{x}}^2 J^{\mathrm{in}}(\boldsymbol{x}, \sigma) = \frac{\sigma - \boldsymbol{x}}{\sigma^3} \odot \exp(-\boldsymbol{x}/\sigma), \tag{16c}$$

$$\nabla_{\boldsymbol{x}, \boldsymbol{x}}^2 J^{\mathrm{in}}(\boldsymbol{x}, \sigma) = \mathrm{diag}(\exp(-\boldsymbol{x}/\sigma))/\sigma^2, \tag{16d}$$

where $\odot$ means element-wise multiplication. Substituting (16) into (14) and let the gradient equals to zero $\frac{dJ^{\mathrm{ex}}}{d\sigma} = 0$, then we have

$$\sigma = \frac{\sum_{i=1}^N x_i^*(\sigma)}{N}. \tag{17}$$

# B    Dataset Description

Our dataset integrates motions from: (i) video-based sources, from which motion data is extracted through our proposed multi-steps motion processing pipeline. The hyperparameters of the pipeline are listed in Table 3; (ii)

open-source datasets: selected motions from AMASS and LAFAN. The dataset comprises 13 distinct motions, which are categorized into three difficulty levels—easy, medium, and hard. To ensure smooth transitions, we linearly interpolate at the beginning and end of each sequence to move from a default pose to the reference motion and back. The details are given in Table 4.

Table 3: Hyperparameters of multi-steps motion processing.

| Hyperparameter | Value |
|---|---|
| $\epsilon_{\text{stab}}$ | 0.1 |
| $\epsilon_{\text{N}}$ | 100 |
| $\epsilon_{\text{vel}}$ | 0.002 |
| $\epsilon_{\text{height}}$ | 0.2 |

Table 4: The details of the highly-dynamic motion dataset.

| Motion name | Motion frames | Source |
|---|---|---|
| **Easy** | | |
| Jabs punch | 285 | video |
| Hooks punch | 175 | video |
| Horse-stance pose | 210 | LAFAN |
| Horse-stance punch | 200 | video |
| **Medium** | | |
| Stretch leg | 320 | video |
| Tai Chi | 500 | video |
| Jump kick | 145 | video |
| Charleston dance | 610 | LAFAN |
| Bruce Lee's pose | 330 | AMASS |
| **Hard** | | |
| Roundhouse kick | 158 | AMASS |
| 360-degree spin | 180 | video |
| Front kick | 155 | video |
| Side kick | 179 | AMASS |

## C   Algorithm Design

### C.1   Observation Space Design

- **Actor observation space:** The actor's observation $s_t^{\text{actor}}$ includes 5-step history of the robot's proprioceptive state $s_t^{\text{prop}}$ and the time-phase variable $\phi_t$.
- **Critic observation space:** The critic's observation $s_t^{\text{crtic}}$ additionally includes the base linear velocity, the body position of the reference motion, the difference between the current and reference body positions, and a set of domain-randomized physical parameters. The details are given in Table 5.

Table 5: Actor and critic observation state space.

| State term | Actor Dim | Critic Dim |
|---|---|---|
| Joint position | $23 \times 5$ | $23 \times 5$ |
| Joint velocity | $23 \times 5$ | $23 \times 5$ |
| Root angular velocity | $3 \times 5$ | $3 \times 5$ |
| Root projected gravity | $3 \times 5$ | $3 \times 5$ |
| Reference motion phase | $1 \times 5$ | $1 \times 5$ |
| Actions | $23 \times 5$ | $23 \times 5$ |
| Root linear velocity | – | $3 \times 5$ |
| Reference body position | – | 81 |
| Body position difference | – | 81 |
| Randomized base CoM offset* | – | 3 |
| Randomized link mass* | – | 22 |
| Randomized stiffness* | – | 23 |
| Randomized damping* | – | 23 |
| Randomized friction coefficient* | – | 1 |
| Randomized control delay* | – | 1 |
| **Total dim** | 380 | 630 |

*Several randomized physical parameters used in domain randomization are part of the critic observation to improve value estimation robustness. The detailed settings of domain randomization are given in Appendix C.3.*

## C.2 Reward Design

All reward functions are detailed in Table 6. Our reward design consists of two main parts: task rewards and regularization rewards. Specifically, we impose penalties when joint position exceeds the soft limits, which are symmetrically scaled from the hard limits by a fixed ratio ($\alpha = 0.95$):

$$\boldsymbol{m} = (\boldsymbol{q}_{\min} + \boldsymbol{q}_{\max})/2, \tag{18a}$$

$$\boldsymbol{d} = \boldsymbol{q}_{\max} - \boldsymbol{q}_{\min}, \tag{18b}$$

$$\boldsymbol{q}_{\text{soft-min}} = \boldsymbol{m} - 0.5 \cdot \boldsymbol{d} \cdot \alpha, \tag{18c}$$

$$\boldsymbol{q}_{\text{soft-max}} = \boldsymbol{m} + 0.5 \cdot \boldsymbol{d} \cdot \alpha, \tag{18d}$$

where $\boldsymbol{q}$ is the joint position. The same procedure is applied to compute the soft limits for joint velocity $\dot{\boldsymbol{q}}$ and torque $\boldsymbol{\tau}$.

Table 6: Reward terms and weights.

| Term | Expression | Weight |
|------|------------|--------|
| **Task** | | |
| Joint position | $\exp(-\|\boldsymbol{q}_t - \hat{\boldsymbol{q}}_t\|_2^2/\sigma_{\text{jpos}})$ | 1.0 |
| Joint velocity | $\exp(-\|\dot{\boldsymbol{q}}_t - \hat{\dot{\boldsymbol{q}}}_t\|_2^2/\sigma_{\text{jvel}})$ | 1.0 |
| Body position | $\exp(-\|\boldsymbol{p}_t - \hat{\boldsymbol{p}}_t\|_2^2/\sigma_{\text{pos}})$ | 1.0 |
| Body rotation | $\exp(-\|\boldsymbol{\theta}_t \ominus \hat{\boldsymbol{\theta}}_t\|_2^2/\sigma_{\text{rot}})$ | 0.5 |
| Body velocity | $\exp(-\|\boldsymbol{v}_t - \hat{\boldsymbol{v}}_t\|_2^2/\sigma_{\text{vel}})$ | 0.5 |
| Body angular velocity | $\exp(-\|\boldsymbol{\omega}_t - \hat{\boldsymbol{\omega}}_t\|_2^2/\sigma_{\text{ang}})$ | 0.5 |
| Body position VR 3 points | $\exp(-\|\boldsymbol{p}_t^{\text{vr}} - \hat{\boldsymbol{p}}_t^{\text{vr}}\|_2^2/\sigma_{\text{pos\_vr}})$ | 1.6 |
| Body position feet | $\exp\left(-\|\boldsymbol{p}_t^{\text{feet}} - \hat{\boldsymbol{p}}_t^{\text{feet}}\|_2^2/\sigma_{\text{pos\_feet}}\right)$ | 1.0 |
| Max Joint position | $\exp\left(-\|\boldsymbol{q}_t - \hat{\boldsymbol{q}}_t\|_\infty / \sigma_{\text{max\_jpos}}\right)$ | 1.0 |
| Contact Mask | $1 - \|c_t - \hat{c}_t\|_1/2$ | 0.5 |
| **Regularization** | | |
| Joint position limits | $\mathbb{I}(\boldsymbol{q} \notin [\boldsymbol{q}_{\text{soft-min}}, \boldsymbol{q}_{\text{soft-max}}])$ | -10.0 |
| Joint velocity limits | $\mathbb{I}(\dot{\boldsymbol{q}} \notin [\dot{\boldsymbol{q}}_{\text{soft-min}}, \dot{\boldsymbol{q}}_{\text{soft-max}}])$ | -5.0 |
| Joint torque limits | $\mathbb{I}(\boldsymbol{\tau} \notin [\boldsymbol{\tau}_{\text{soft-min}}, \boldsymbol{\tau}_{\text{soft-max}}])$ | -5.0 |
| Slippage | $\|\boldsymbol{v}_{\text{xy}}^{\text{feet}}\|_2^2 \cdot \mathbb{I}[\|F^{\text{feet}}\|_2 \geq 1]$ | -1.0 |
| Feet contact forces | $\min(\|F^{\text{feet}} - 400\|_2^2, 0)$ | -0.01 |
| Feet air time[30] | $\mathbb{I}[T_{\text{air}} > 0.3]$ | -1.0 |
| Stumble | $\mathbb{I}[\|\boldsymbol{F}_{xy}^{\text{feet}}\| > 5 \cdot F_z^{\text{feet}}]$ | -2.0 |
| Torque | $\|\boldsymbol{\tau}\|_2^2$ | -1e-6 |
| Action rate | $\|\boldsymbol{a}_t - \boldsymbol{a}_{t-1}\|_2^2$ | -0.02 |
| Collision | $\mathbb{I}_{\text{collision}}$ | -30 |
| Termination | $\mathbb{I}_{\text{termination}}$ | -200 |

## C.3 Domain Randomization

To improve the transferability of our trained polices to real-world settings, we incorporate domain randomization during training to support robust sim-to-sim and sim-to-real transfer. The specific settings are given in Table 7.

## C.4 PPO Hyperparameter

The detailed PPO hyperparameters are shown in Table 8.

Table 7: Domain randomization settings.

| Term | Value |
|------|-------|
| Dynamics Randomization | |
| Friction | $\mathcal{U}(0.2,\ 1.2)$ |
| PD gain | $\mathcal{U}(0.9,\ 1.1)$ |
| Link mass(kg) | $\mathcal{U}(0.9,\ 1.1)\times$ default |
| Ankle inertia(kg·m$^2$) | $\mathcal{U}(0.9,\ 1.1)\times$ default |
| Base CoM offset(m) | $\mathcal{U}(-0.05,\ 0.05)$ |
| ERFI[58](N·m/kg) | $0.05\times$ torque limit |
| Control delay(ms) | $\mathcal{U}(0,\ 40)$ |
| External Perturbation | |
| Random push interval(s) | $[5,\ 10]$ |
| Random push velocity(m/s) | 0.1 |

Table 8: Hyperparameters related to PPO.

| Hyperparameter | Value |
|----------------|-------|
| Optimizer | Adam |
| Batch size | 4096 |
| Mini Batches | 4 |
| Learning epoches | 5 |
| Entropy coefficient | 0.01 |
| Value loss coefficient | 1.0 |
| Clip param | 0.2 |
| Max grad norm | 1.0 |
| Init noise std | 0.8 |
| Learning rate | 1e-3 |
| Desired KL | 0.01 |
| GAE decay factor($\lambda$) | 0.95 |
| GAE discount factor($\gamma$) | 0.99 |
| Actor MLP size | [512, 256, 128] |
| Critic MLP size | [768, 512, 128] |
| MLP Activation | ELU |

## C.5 Curriculum Learning

To imitate high-dynamic motions, we introduce two curriculum mechanisms: a termination curriculum that gradually reduces tracking error tolerance, and a penalty curriculum that progressively increases the weight of regularization terms, promoting more stable and physically plausible behaviors.

- **Termination Curriculum**: The episode is terminated early when the humanoid's motion deviates from the reference beyond a termination threshold $\theta$. During training, this threshold is gradually decreased to increase the difficulty:

$$\theta \leftarrow \text{clip}\left(\theta \cdot (1 - \delta),\ \theta_{\min},\ \theta_{\max}\right), \tag{19}$$

  where the initial threshold $\theta = 1.5$, with bounds $\theta_{\min} = 0.3$, $\theta_{\max} = 2.0$, and decay rate $\delta = 2.5 \times 10^{-5}$.

- **Penalty Curriculum**: To facilitate learning in the early training stages while gradually enforcing stronger regularization, we introduce a scaling factor $\alpha$ that increases progressively to modulate the influence of the penalty term:

$$\alpha \leftarrow \text{clip}\left(\alpha \cdot (1 + \delta),\ \alpha_{\min},\ \alpha_{\max}\right), \quad \hat{r}_{\text{penalty}} \leftarrow \alpha \cdot r_{\text{penalty}}, \tag{20}$$

  where the initial penalty scale $\alpha = 0.1$, with bounds $\alpha_{\min} = 0.0$, $\alpha_{\max} = 1.0$, and growth rate $\delta = 1.0 \times 10^{-4}$.

## C.6 PD Controller Parameter

The gains of the PD controller are listed in Table 9. To improve the numerical stability and fidelity of the simulator in training, we manually set the inertia of the ankle links to a fixed value of $5 \times 10^{-3}$.

Table 9: PD controller gains.

| Joint name | Stiffness ($k_p$) | Damping ($k_d$) |
|------------|-------------------|-----------------|
| Left/right shoulder pitch/roll/yaw | 100 | 2.0 |
| Left/right shoulder yaw | 50 | 2.0 |
| Left/right elbow | 50 | 2.0 |
| Waist pitch/roll/yaw | 400 | 5.0 |
| Left/right hip pitch/roll/yaw | 100 | 2.0 |
| Left/right knee | 150 | 4.0 |
| Left/right ankle pitch/roll | 40 | 2.0 |

# D Experimental Details

## D.1 Experiment Setup

- **Compute platform**: Each experiment is conducted on a machine with a 24-core Intel i7-13700 CPU running at 5.2GHz, 32 GB of RAM, and a single NVIDIA GeForce RTX 4090 GPU, with Ubuntu 20.04. Each of our models is trained for 27 hours.

- **Real robot setup**: We deploy our policies on a Unitree G1 robot. The system consists of an onboard motion control board and an external PC, connected via Ethernet. The control board collects sensor data and transmits it to the PC using the DDS protocol. The PC maintains observation history, performs policy inference, and sends target joint angles back to the control board, which then issues motor commands.

## D.2 Evaluation Metrics

- Global Mean Per Body Position Error ($E_{\text{g-mpbpe}}$, mm): The average position error of body parts in global coordinates.

$$E_{\text{g-mpbpe}} = \mathbb{E}\left[\left\|\boldsymbol{p}_t - \boldsymbol{p}_t^{\text{ref}}\right\|_2\right]. \tag{21}$$

- Root-Relative Mean Per Body Position Error ($E_{\text{mpbpe}}$, mm): The average position error of body parts relative to the root position.

$$E_{\text{mpbpe}} = \mathbb{E}\left[\left\|(\boldsymbol{p}_t - \boldsymbol{p}_{\text{root,t}}) - (\boldsymbol{p}_t^{\text{ref}} - \boldsymbol{p}_{\text{root,t}}^{\text{ref}})\right\|_2\right]. \tag{22}$$

- Mean Per Joint Position Error ($E_{\text{mpjpe}}$, $10^{-3}$ rad): The average angular error of joint rotations.

$$E_{\text{mpjpe}} = \mathbb{E}\left[\left\|\boldsymbol{q}_t - \boldsymbol{q}_t^{\text{ref}}\right\|_2\right]. \tag{23}$$

- Mean Per Joint Velocity Error ($E_{\text{mpjve}}$, $10^{-3}$ rad/frame): The average error of joint angular velocities.

$$E_{\text{mpjve}} = \mathbb{E}\left[\left\|\Delta\boldsymbol{q}_t - \Delta\boldsymbol{q}_t^{\text{ref}}\right\|_2\right], \tag{24}$$

where $\Delta\boldsymbol{q}_t = \boldsymbol{q}_t - \boldsymbol{q}_{t-1}$.

- Mean Per Body Velocity Error ($E_{\text{mpbve}}$, mm/frame): The average error of body part linear velocities.

$$E_{\text{mpbve}} = \mathbb{E}\left[\left\|\Delta\boldsymbol{p}_t - \Delta\boldsymbol{p}_t^{\text{ref}}\right\|_2\right], \tag{25}$$

where $\Delta\boldsymbol{p}_t = \boldsymbol{p}_t - \boldsymbol{p}_{t-1}$.

- Mean Per Body Acceleration Error ($E_{\text{mpbae}}$, mm/frame²): The average error of body part accelerations.

$$E_{\text{mpbae}} = \mathbb{E}\left[\left\|\Delta^2\boldsymbol{p}_t - \Delta^2\boldsymbol{p}_t^{\text{ref}}\right\|_2\right], \tag{26}$$

where $\Delta^2\boldsymbol{p}_t = \Delta\boldsymbol{p}_t - \Delta\boldsymbol{p}_{t-1}$.

## D.3 Baseline Implementations

To ensure fair comparison, all baseline methods are trained separately for each motion. We consider the following baselines:

- **OmniH2O**: OmniH2O adopts a teacher-student training paradigm. We moderately increase the tracking reward weights to better match the G1 robot. In our setup, the teacher and student policies are trained for 20 and 10 hours, respectively.

- **Exbody2**: ExBody2 utilizes a decoupled keypoint-velocity tracking mechanism. The teacher and student policies are trained for 20 and 10 hours, respectively.

- **MaskedMimic**: MaskedMimic comprises three sequential training phases and we utilize only the first phase, as the remaining stages are not pertinent to our tasks. The method focuses on reproducing reference motions by directly optimizing pose-level accuracy, without explicit regularization of physical plausibility. Each policy is trained for 18 hours.

### D.4 Tracking Factor Configurations

We define five sets of tracking factors: Coarse, Medium, UpperBound, LowerBound, and the initial values of Ours, as shown in Table 10. We also provide the converged tracking factors of our adaptive mechanism in Table 11.

Table 10: Tracking factors in different configurations.

| Factor term | Ours(Init) | Coarse | Medium | Upperbound | Lowerbound |
|---|---|---|---|---|---|
| Joint position | 0.3 | 0.3 | 0.1 | 0.08 | 0.02 |
| Joint velocity | 30.0 | 30.0 | 10.0 | 5.0 | 2.5 |
| Body position | 0.015 | 0.015 | 0.005 | 0.002 | 0.0003 |
| Body rotation | 0.1 | 0.1 | 0.03 | 0.4 | 0.02 |
| Body velocity | 1.0 | 1.0 | 0.3 | 0.12 | 0.03 |
| Body angular velocity | 15.0 | 15.0 | 5.0 | 3.0 | 1.5 |
| Body position VRpoints | 0.015 | 0.015 | 0.005 | 0.003 | 0.0003 |
| Body position feet | 0.01 | 0.01 | 0.003 | 0.003 | 0.0002 |
| Max joint position | 1.0 | 1.0 | 0.3 | 0.5 | 0.25 |

Table 11: Converged tracking factors of our adaptive mechanism across different motions in the ablation study of Section 5.4.

| Factor term | Jabs punch | Charleston dance | Bruce Lee's pose | Roundhouse kick |
|---|---|---|---|---|
| Joint position | $0.0310 \pm 0.0002$ | $0.0360 \pm 0.0016$ | $0.0268 \pm 0.0009$ | $0.0261 \pm 0.0005$ |
| Joint velocity | $2.8505 \pm 0.0419$ | $5.5965 \pm 0.1797$ | $3.6053 \pm 0.0323$ | $4.3859 \pm 0.0537$ |
| Body position | $0.0007 \pm 0.0000$ | $0.0023 \pm 0.0001$ | $0.0025 \pm 0.0000$ | $0.0010 \pm 0.0000$ |
| Body rotation | $0.0998 \pm 0.0000$ | $0.0544 \pm 0.0016$ | $0.0046 \pm 0.0001$ | $0.0829 \pm 0.0176$ |
| Body velocity | $0.0554 \pm 0.0006$ | $0.0941 \pm 0.0013$ | $0.0768 \pm 0.0001$ | $0.0929 \pm 0.0008$ |
| Body angular velocity | $1.8063 \pm 0.0076$ | $2.8267 \pm 0.0841$ | $2.1706 \pm 0.0050$ | $3.0238 \pm 0.0303$ |
| Body position VRpoints | $0.0008 \pm 0.0000$ | $0.0031 \pm 0.0002$ | $0.0024 \pm 0.0000$ | $0.0015 \pm 0.0000$ |
| Body position feet | $0.0006 \pm 0.0000$ | $0.0031 \pm 0.0001$ | $0.0028 \pm 0.0000$ | $0.0011 \pm 0.0000$ |
| Max joint position | $0.3963 \pm 0.0003$ | $0.4339 \pm 0.0124$ | $0.3299 \pm 0.0064$ | $0.3352 \pm 0.0010$ |

# E  Additional Experimental Results

## E.1  Analysis of Contact Mask Estimation and Motion Correction Method

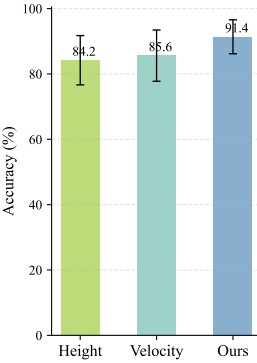

Figure 10: Accuracy of contact mask estimation across different methods.

Fig. 10 illustrates the accuracy of the proposed contact mask estimation method, evaluated on a manually labeled motion dataset with 10 samples. The proposed approach demonstrates an impressive accuracy of 91.4%.

Fig. 11 presents a visual comparison of the efficacy of the proposed motion correction technique in mitigating floating artifacts. Prior to motion correction, the overall height of the SMPL model is noticeably elevated relative to the ground level. In contrast, after applying the correction, the model's motion aligns more accurately with the ground plane, effectively reducing the observed floating artifacts.

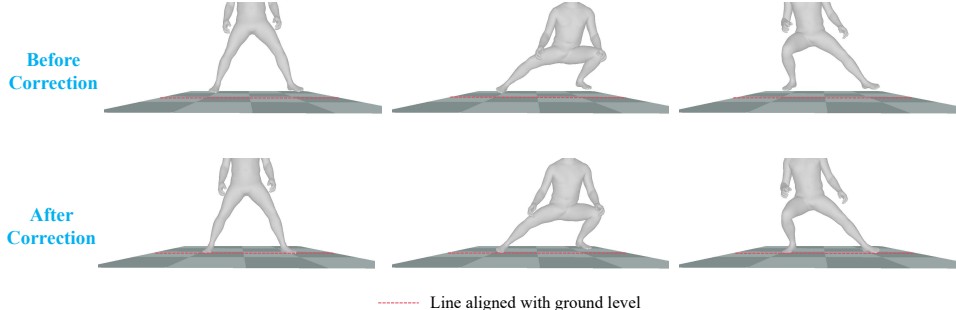

Before Correction

After Correction

------- Line aligned with ground level

Figure 11: Visualization of motion correction effectiveness in mitigating floating artifacts.

## E.2 Ablation Study of Adaptive Motion Tracking Mechanism

Table 12 presents the ablation study results evaluating the impact of different tracking factors on four motion tasks: Jabs Punch, Charleston Dance, Roundhouse Kick, and Bruce Lee's Pose.

Table 12: Ablation results of adaptive motion tracking mechanism in Section 5.4.

| Method | $E_{\text{g-mpbpe}} \downarrow$ | $E_{\text{mpbpe}} \downarrow$ | $E_{\text{mpjpe}} \downarrow$ | $E_{\text{mpbve}} \downarrow$ | $E_{\text{mpbae}} \downarrow$ | $E_{\text{mpjve}} \downarrow$ |
|---|---|---|---|---|---|---|
| Jabs punch | | | | | | |
| Ours | $\mathbf{44.38}_{\pm 7.118}$ | $\mathbf{28.00}_{\pm 3.533}$ | $783.36_{\pm 11.73}$ | $5.52_{\pm 0.156}$ | $\mathbf{6.23}_{\pm 0.063}$ | $88.01_{\pm 2.465}$ |
| Coarse | $63.95_{\pm 6.680}$ | $36.76_{\pm 2.743}$ | $921.50_{\pm 16.70}$ | $6.16_{\pm 0.011}$ | $6.46_{\pm 0.042}$ | $91.46_{\pm 0.465}$ |
| Medium | $51.07_{\pm 2.635}$ | $30.93_{\pm 2.635}$ | $790.54_{\pm 22.82}$ | $5.68_{\pm 0.140}$ | $6.31_{\pm 0.057}$ | $90.19_{\pm 1.821}$ |
| Upperbound | $45.74_{\pm 1.702}$ | $28.72_{\pm 1.702}$ | $793.52_{\pm 8.888}$ | $\mathbf{5.43}_{\pm 0.066}$ | $6.29_{\pm 0.085}$ | $88.68_{\pm 0.727}$ |
| Lowerbound | $48.66_{\pm 0.488}$ | $28.97_{\pm 0.487}$ | $\mathbf{781.73}_{\pm 16.72}$ | $5.61_{\pm 0.079}$ | $6.31_{\pm 0.026}$ | $88.44_{\pm 1.397}$ |
| Charleston dance | | | | | | |
| Ours | $94.81_{\pm 14.18}$ | $43.09_{\pm 5.748}$ | $886.91_{\pm 74.76}$ | $\mathbf{6.83}_{\pm 0.346}$ | $7.26_{\pm 0.034}$ | $162.70_{\pm 7.133}$ |
| Coarse | $119.24_{\pm 4.501}$ | $55.80_{\pm 1.324}$ | $1288.02_{\pm 3.807}$ | $7.54_{\pm 0.180}$ | $7.28_{\pm 0.021}$ | $178.61_{\pm 3.304}$ |
| Medium | $\mathbf{83.63}_{\pm 3.159}$ | $\mathbf{41.02}_{\pm 1.743}$ | $\mathbf{933.33}_{\pm 38.23}$ | $6.89_{\pm 0.185}$ | $\mathbf{7.22}_{\pm 0.011}$ | $\mathbf{164.92}_{\pm 4.380}$ |
| Upperbound | $86.90_{\pm 8.651}$ | $41.92_{\pm 2.632}$ | $917.64_{\pm 14.85}$ | $7.02_{\pm 0.103}$ | $7.22_{\pm 0.041}$ | $167.64_{\pm 1.089}$ |
| Lowerbound | $358.82_{\pm 10.35}$ | $145.42_{\pm 1.109}$ | $1199.21_{\pm 12.78}$ | $8.99_{\pm 0.050}$ | $8.48_{\pm 0.033}$ | $167.25_{\pm 0.783}$ |
| Roundhouse kick | | | | | | |
| Ours | $\mathbf{52.53}_{\pm 2.106}$ | $\mathbf{28.39}_{\pm 1.400}$ | $708.55_{\pm 16.04}$ | $6.85_{\pm 0.196}$ | $7.13_{\pm 0.046}$ | $106.22_{\pm 0.715}$ |
| Coarse | $76.81_{\pm 2.863}$ | $38.98_{\pm 2.230}$ | $1008.32_{\pm 29.74}$ | $7.49_{\pm 0.234}$ | $7.57_{\pm 0.044}$ | $108.40_{\pm 0.010}$ |
| Medium | $63.12_{\pm 5.178}$ | $33.74_{\pm 2.336}$ | $806.84_{\pm 66.23}$ | $7.03_{\pm 0.125}$ | $7.32_{\pm 0.046}$ | $104.77_{\pm 1.319}$ |
| Upperbound | $54.95_{\pm 2.164}$ | $31.31_{\pm 0.344}$ | $766.32_{\pm 12.92}$ | $\mathbf{6.93}_{\pm 0.013}$ | $7.19_{\pm 0.012}$ | $105.64_{\pm 1.911}$ |
| Lowerbound | $70.10_{\pm 2.674}$ | $36.29_{\pm 1.475}$ | $\mathbf{715.01}_{\pm 34.01}$ | $7.08_{\pm 0.102}$ | $7.32_{\pm 0.067}$ | $\mathbf{102.50}_{\pm 4.650}$ |
| Bruce Lee's pose | | | | | | |
| Ours | $196.22_{\pm 17.03}$ | $69.12_{\pm 2.392}$ | $972.04_{\pm 49.27}$ | $7.57_{\pm 0.214}$ | $8.54_{\pm 0.198}$ | $94.36_{\pm 3.750}$ |
| Coarse | $239.06_{\pm 51.74}$ | $80.78_{\pm 15.81}$ | $1678.34_{\pm 394.3}$ | $8.42_{\pm 0.525}$ | $8.93_{\pm 0.422}$ | $112.30_{\pm 10.87}$ |
| Medium | $470.24_{\pm 249.2}$ | $206.92_{\pm 116.1}$ | $4490.80_{\pm 105.1}$ | $9.58_{\pm 0.085}$ | $9.61_{\pm 0.080}$ | $99.65_{\pm 2.441}$ |
| Upperbound | $250.64_{\pm 178.6}$ | $93.70_{\pm 65.09}$ | $1358.02_{\pm 561.6}$ | $8.31_{\pm 2.160}$ | $8.94_{\pm 1.384}$ | $106.30_{\pm 23.06}$ |
| Lowerbound | $\mathbf{158.12}_{\pm 2.934}$ | $\mathbf{60.54}_{\pm 1.554}$ | $\mathbf{955.10}_{\pm 37.04}$ | $\mathbf{7.05}_{\pm 0.040}$ | $\mathbf{7.94}_{\pm 0.051}$ | $\mathbf{81.60}_{\pm 1.277}$ |

## E.3 Ablation Study of Contact Mask

To evaluate the effectiveness of the contact mask, we additionally conducted an ablation study on three representative motions characterized by distinct foot contact patterns: Charleston Dance, Jump Kick, and Roundhouse Kick. We additionally introduce the mean foot contact mask error as a metric:

$$E_{\text{contact-mask}} = \mathbb{E}\left[ \|c_t - \hat{c}_t\|_1 \right]. \tag{27}$$

The results, shown in Table 13, demonstrate that our method significantly reduces foot contact errors $E_{\text{contact-mask}}$ compared to the baseline without the contact mask. In addition, it also leads to noticeable improvements in other tracking metrics, validating the effectiveness of the proposed contact-aware design.

## E.4 Additional Real-World Results

Fig. 12 presents additional results of deploying our policy in the real world, covering more highly-dynamic motions. These results further validate the effectiveness of our method in tracking high-dynamic motions, enabling the humanoid to learn more expressive skills.

Table 13: Ablation results of contact mask.

| Method | $E_{\text{contact-mask}} \downarrow$ | $E_{\text{mpbpe}} \downarrow$ | $E_{\text{mpjpe}} \downarrow$ | $E_{\text{mpbve}} \downarrow$ | $E_{\text{mpbae}} \downarrow$ |
|---|---|---|---|---|---|
| Charleston dance | | | | | |
| Ours | $217.82_{\pm47.97}$ | $43.09_{\pm5.748}$ | $886.91_{\pm74.76}$ | $6.83_{\pm0.346}$ | $7.26_{\pm0.034}$ |
| Ours w/o contact mask | $633.91_{\pm49.74}$ | $76.13_{\pm53.01}$ | $980.40_{\pm222.0}$ | $7.72_{\pm1.439}$ | $7.64_{\pm0.594}$ |
| Jump kick | | | | | |
| Ours | $294.22_{\pm6.037}$ | $42.58_{\pm8.126}$ | $840.33_{\pm97.76}$ | $9.48_{\pm0.717}$ | $10.21_{\pm10.21}$ |
| Ours w/o contact mask | $386.75_{\pm6.036}$ | $170.28_{\pm97.29}$ | $1259.21_{\pm423.9}$ | $16.92_{\pm0.012}$ | $16.57_{\pm5.810}$ |
| Roundhouse kick | | | | | |
| Ours | $243.16_{\pm1.778}$ | $28.39_{\pm1.400}$ | $708.55_{\pm16.04}$ | $6.85_{\pm0.196}$ | $7.33_{\pm0.046}$ |
| Ours w/o contact mask | $250.10_{\pm6.123}$ | $36.76_{\pm2.743}$ | $921.52_{\pm16.70}$ | $6.16_{\pm0.012}$ | $6.46_{\pm0.042}$ |

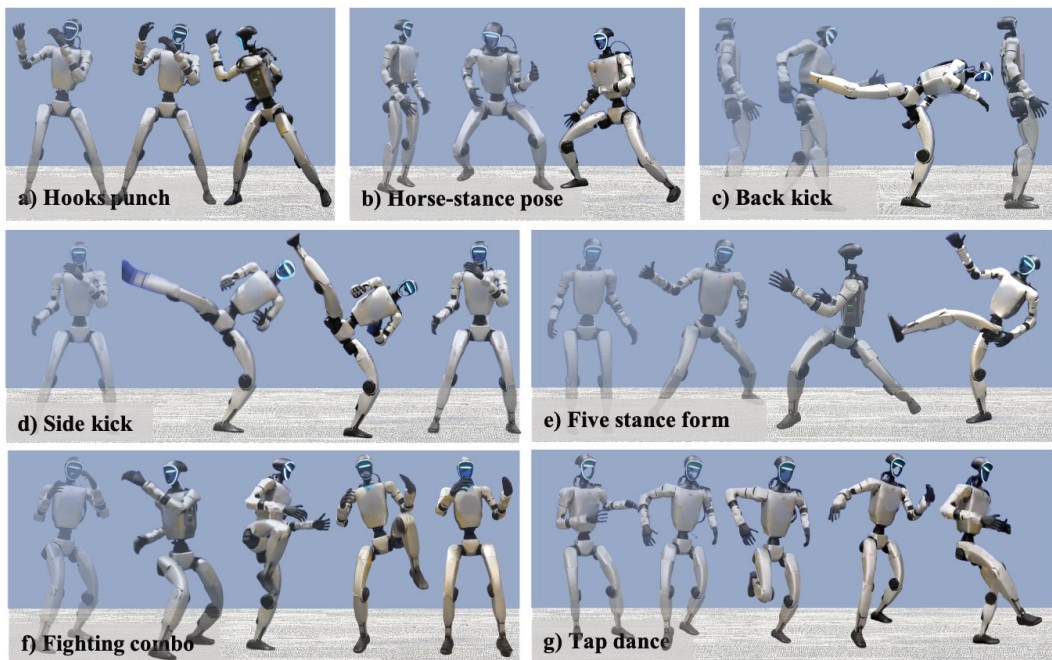

Figure 12: Our robot masters more dynamic skills in the real world. Time flows left to right.

# F  Broader Impact

Our work advances humanoid robotics by enabling the imitation of complex, highly-dynamic human motions such as martial arts and dancing. This has broad potential in fields like physical assistance, rehabilitation, education, and entertainment, where expressive and agile robot behavior can support training, therapy, and interactive experiences. However, such capabilities also raise important ethical and societal concerns. High-agility robots interacting closely with humans introduce safety risks, and their potential to replace skilled human roles in performance, instruction, or service contexts may lead to labor displacement. Moreover, the misuse of advanced motion imitation—for example, in surveillance or military applications—poses security concerns. These risks call for clear regulation, strong safety mechanisms, and human oversight. Additionally, the environmental cost of training models and operating physical robots highlights the need for energy-efficient and sustainable development. We believe this work should be viewed as a step toward responsible, human-aligned robotics, and we encourage continued dialogue on its societal impact.

