# OpenReview forum: "KungfuBot: Physics-Based Humanoid Whole-Body Control for Learning Highly-Dynamic Skills"
_NeurIPS.cc/2025/Conference — NeurIPS 2025 poster_

### Official Review · Reviewer_339U · 2025-06-06

**Clarity:** 3
**Significance:** 2
**Originality:** 3
**Rating:** 3
**Confidence:** 5

**Summary:**

The paper introduces KungfuBot, a framework for learning highly dynamic whole-body humanoid control using reinforcement learning. It tackles the limitations of prior approaches that struggle with fast, agile motions by combining a multi-stage physics-based motion processing pipeline with an adaptive motion tracking mechanism. The system extracts human motions from video, filters them using physical feasibility criteria, corrects artifacts, and retargets them to a humanoid robot. A novel bi-level optimization formulation dynamically adjusts tracking precision during training, enabling better reward shaping. The policy is trained using an asymmetric actor-critic RL architecture and successfully deployed zero-shot on the Unitree G1 robot, demonstrating expressive skills such as Kungfu and dancing. The method significantly outperforms existing baselines in both simulation and real-world evaluations.

**Questions:**

Please address the weaknesses raised above.

**Ethical Concerns:**

["NO or VERY MINOR ethics concerns only"]

**Final Justification:**

Most of my concerns have been addressed by the authors during the rebuttal phase. Given the work's limited improvement versus existing works and the missing details as pointed out, I will increase my score to borderline reject.

**Limitations:**

Yes.

**Paper Formatting Concerns:**

No.

**Quality:**

3

**Strengths And Weaknesses:**

**Strengths**
- The proposed method enables a humanoid robot to learn and execute agile and complex motions (e.g., Kungfu, dance), outperforming prior work in tracking precision and expressiveness.
- The pipeline uses physical plausibility metrics (CoM-CoP stability) to filter out untrackable motions, increasing training efficiency and real-world robustness.
- The introduction of a principled, adaptive reward shaping mechanism that dynamically tunes tracking precision is both novel and practically impactful, helping generalize across motions of varying difficulty.
- The real-world experiments look strong and convincing.


**Weaknesses**
- It might not be justified to compare the method with ASAP on the number of stages if each stage of the proposed method involves multiple steps. Additionally, the MoCap component in ASAP is used to propose delta actions as real-time compensations, which deal with a topic that the paper does not cover.
- In motion filtering, it is not clear how the correct thresholds for stability, velocity and height are selected. Experiment 5.2 justifies the selection with policies trained on each of the motions. However, considering the inherent difficulty within the motions, some of the learnable but hard ones are filtered out depending on these base policies.
- If motion filtering is presented as a contribution of the paper, it is worth considering discussions about other methods such as progressive training [1], learning-progress based automatic curriculum [2], physics-based retargetting [3].
- How is the formulation of the adaptive tracking factor tuning related to a formulation based on a constrained MDP, where the objective is the accumulated negative tracking error and the constraint is the internal objective and the regularizations being larger than some threshold?
- The approximation of the average optimal tracking error with the instantaneous tracking error is not reasonable, as it boils down to a simple curriculum where $\sigma$ decreases (which tightens the distance) when $\hat{x}$ goes down (error becomes smaller).
- Each policy is trained for a specific motion. While the proposed adaptive tracking strategy demonstrates effectiveness, the paper fails to show a further generality to multiple motions. I would imagine the tracking factor would also vary across different motions in that case.
- As each policy is trained for a specific motion and the selected baselines are focusing on multi-motion tracking, whose performance might be hindered due to this ambitious objective, it may make more sense to evaluate the proposed method against simple baselines on tracking single motions, such as DeepMimic.


[1] Luo, Z., Cao, J., Kitani, K. and Xu, W., 2023. Perpetual humanoid control for real-time simulated avatars. In Proceedings of the IEEE/CVF International Conference on Computer Vision (pp. 10895-10904).

[2] Li, C., Stanger-Jones, E., Heim, S. and Kim, S., 2024. Fld: Fourier latent dynamics for structured motion representation and learning. arXiv preprint arXiv:2402.13820.

[3] Grandia, R., Farshidian, F., Knoop, E., Schumacher, C., Hutter, M. and Bächer, M., 2023. Doc: Differentiable optimal control for retargeting motions onto legged robots. ACM Transactions on Graphics (TOG), 42(4), pp.1-14.

---

> ### Author Rebuttal · Authors · 2025-07-29
>
> Thanks for the detailed and thoughtful feedback. We are encouraged that reviewer 339U appreciated our framework’s ability to learn expressive and agile humanoid skills and the strength of our real-world results. We address the raised concerns in detail below.
>
>
> # W1: ASAP
> We agree that comparing the number of training stages with ASAP was imprecise. We will revise the text to avoid this unfair comparison. Additionally, we acknowledge that ASAP’s delta action policy addresses sim-to-real compensation in a different way, which is beyond the scope of our method. We will clarify this distinction in the paper.
>
> # W2: Threshold selection in motion filter
>
> Our goal in motion filtering is not to establish an absolute notion of trackability, but rather to reliably distinguish clearly learnable motions from those that consistently fail under practical training settings. While some borderline cases might eventually succeed with more sophisticated methods, our thresholds serve the practical goal of enabling robust policy training at scale.
>
> We selected the filtering thresholds based on small-scale preliminary experiments and then validated them on a larger set. To further justify the selected parameters, we provide ablations below. Each table sweeps one parameter with others fixed to default (asterisk, \*) values. The results support the effectiveness of our chosen thresholds.
>
> For the motion filtering thresholds, we report the F1 score $F_1= \frac{2*\text{Precision}\cdot{\text{Recall}}}{\text{Precision}+{\text{Recall}}}$ of classifying motions as trackable (Episode Length Ratio > 60%) on the same dataset as in Section 5.2, where $\epsilon_{\mathrm{stab}}$ defines the maximum CoM–CoP distance for a frame to be considered stable, and $\epsilon_N$ sets the maximum length of consecutive unstable frames allowed in a motion.
>
> | $\epsilon_{stab}$ | 0.01   | 0.05   | 0.1\* | 0.2    | 0.5  |
> |:---------------:|--------|--------|-----|--------|-----|
> |     F1 Score $↑$    | 0.5714 | 0.6154 | *1*\*   | 0.6667 | 0    |
>
> | $\epsilon_N$ | 10     | 50     | 100\* | 150    | 200    |
> |:------------:|--------|--------|-----|--------|--------|
> |   F1 Score $↑$  | 0.6154 | 0.6154 | *1*\*   | 0.8571 | 0.6667 |
>
>
>
> For the thresholds used in contact mask estimation, we evaluate the accuracy of correctly detecting both feet’s contact states, where the contact state of one feet is estimated according to $c_t=\mathbb{I}[\|\|p_{t+1}-p_t \|\|^2_2<\epsilon_{vel}]\cdot \mathbb{I}[ p_{t,z}<\epsilon_{height} ]$. The results are obtained from the same dataset as Appendix E.1 (Fig 9).
>
>
> | $\epsilon_{vel}$ | 0.0001           | 0.001            | 0.002\*            | 0.003            | 0.005            | 0.01             |
> |--------------|------------------|------------------|------------------|------------------|------------------|------------------|
> |        Acc(%)$↑$     | 77.42 $\pm$ 8.65 | 90.82 $\pm$ 4.51 | *91.40 $\pm$ 5.10*\* | 89.72 $\pm$ 5.28 | 88.37 $\pm$ 5.42 | 87.00 $\pm$ 5.80 |
>
> | $\epsilon_{height}$ | 0.01     | 0.1         | 0.15             | 0.2\*              | 0.25             | 0.3              | 0.5              |
> |:-----------------:|------------------|-------------------|------------------|------------------|------------------|------------------|------------------|
> |        Acc(%)$↑$        | 7.59 $\pm$ 15.19 | 11.21 $\pm$ 16.31 | 81.33 $\pm$ 9.41 | *91.40 $\pm$ 5.10*\* | 87.93 $\pm$ 6.00 | 86.90 $\pm$ 6.20 | 84.87 $\pm$ 7.35 |
>
>
> # W3: Discussion about other motion filter methods
>
> Thank you for the suggestion. We will add a discussion comparing our motion filtering with related approaches. Our method focuses on physically-plausible motion selection prior to training, aiming to reduce downstream learning difficulty. In contrast, progressive training and learning-progress curricula adaptively adjust training difficulty during learning. DOC retargets motions via differentiable optimization to obtain physically feasible motion, but has so far only succeeded on quadruped robots due to the complexity of humanoid dynamics.
>
>
> # W4: Formulation of adaptive tracking factor
>
> While we did not explicitly formulate our method as a constrained MDP, the goal is indeed related—both aim to optimize final policy performance under certain constraints.
>
> Our bi-level formulation separates concerns: the lower level corresponds a standard RL problem (MDP), while the upper level considers the tracking factor based on absolute policy performance. This separation makes our method compatible with off-the-shelf RL algorithms.
>
> A constrained MDP formulation could express similar objectives by integrating external and internal objectives. While the formulation is more unified, solving CMDPs in practice is more complex and often requires custom algorithms such as Lagrangian relaxation or dual ascent.
>
>
>
> # W5: Reasonability of adaptive curriculum
>
> We approximate the average tracking error using an *exponential moving average* of the instantaneous error during training. EMA provides a smooth and relatively stable estimate of the current policy’s average tracking error, and it gradually approaches the optimal tracking error as the policy converges.
>
> Although the resulting update rule for sigma is simple in form, it is grounded in a principled formulation derived from our bi-level optimization. To our knowledge, ours is the first to provide a quantitative rule for determining sigma in motion tracking, supported by theoretical and empirical insights.
>
>
> # W6: Generality to multiple motions
>
> Our formulation can be naturally extended to the multi-motion setting by concatenating the error sequences from all motions into a single sequence. The rest of the derivation remains unchanged.
>
> We acknowledge that different motions may have different optimal tracking factors. In multi-motion training, however, a unified tracking factor is typically required, as in prior works such as OmniH2O \[1\], Exbody2 \[2\] and GMT \[3\]. In such cases, our formulation has the potential to offer an advantage by adaptively estimating a shared tracking factor, which may outperform manually tuned fixed values.
>
> We focused on single-motion training in this work to better study highly dynamic and expressive motions. Extending the method to multi-motion training is a promising direction that we believe deserves a dedicated investigation.
>
> \[1\] T. He _et al._, “OmniH2O: Universal and Dexterous Human-to-Humanoid Whole-Body Teleoperation and Learning,” Jun. 13, 2024, _arXiv_: arXiv:2406.08858.
>
> \[2\] M. Ji _et al._, “ExBody2: Advanced Expressive Humanoid Whole-Body Control,” Mar. 12, 2025, _arXiv_: arXiv:2412.13196.
>
> \[3\] Z. Chen _et al._, “GMT: General Motion Tracking for Humanoid Whole-Body Control,” Jun. 17, 2025, _arXiv_: arXiv:2506.14770.
>
>
> # W7: Baseline on single-motion tracking
>
> We clarify that all baselines, including OmniH2O and Exbody2, are trained individually for each motion, rather than using a multi-motion training setup, therefore ensures a fair comparison. Our MaskedMimic baseline in fact corresponds to the first-stage policy of the original paper only, which is equivalent to DeepMimic you mentioned.
>
> Additionally, we conducted additional ablation experiments to enable a fairer comparison with MaskedMimic’s oracle setting. Specifically, we trained our method with privileged information and without domain randomization, which we refer to as *Ours (Oracle)*. The results are presented in the table below.
>
>
> It demonstrates that our method achieves competitive or superior performance to MaskedMimic under identical oracle settings.
>
> |  Level   | Method         | $E_{g-mpbpe}↓$    | $E_{mpbpe}↓$      | $E_{mpjpe}↓$       | $E_{mpbve}↓$     | $E_{mpbae}↓$     | $E_{mpjve}↓$       |
> | --- | -------------- | ---------------- | ---------------- | ----------------- | --------------- | --------------- | ----------------- |
> |  Easy   | Ours           | 53.25 $\pm$ 8.80 | 28.16 $\pm$ 3.06 | 725.62 $\pm$ 8.10 | 4.41 $\pm$ 0.16 | 4.65 $\pm$ 0.07 | 81.28 $\pm$ 1.02  |
> |     | MaskedMimic （Oracle）   | 41.79 $\pm$ 0.86 | 21.86 $\pm$ 1.02 | 739.96 $\pm$ 9.98 | 5.20 $\pm$ 0.12 | 7.40 $\pm$ 0.17 | 132.01 $\pm$ 4.47 |
> |     | **Ours（Oracle）** |   45.02 $\pm$ 3.38 | 22.95 $\pm$ 7.61 | 710.30 $\pm$ 8.33 | 4.63 $\pm$ 0.79 | 4.89 $\pm$ 0.48 | 73.44$\pm$ 6.21 |
> |  Medium   | Ours           | 126.48 $\pm$ 12.08 | 48.87 $\pm$ 3.38  | 1043.30 $\pm$ 46.67 | 6.62 $\pm$ 0.18 | 7.19 $\pm$ 0.11  | 105.30 $\pm$ 2.66  |
> |     | MaskedMimic（Oracle）    | 150.92 $\pm$ 59.65 | 61.69 $\pm$ 20.58 | 934.25 $\pm$ 69.35  | 8.16 $\pm$ 0.88 | 10.01 $\pm$ 0.39 | 176.84 $\pm$ 11.68 |
> |     | **Ours（Oracle）** |       66.85 $\pm$ 22.49 | 29.56 $\pm$ 6.50 | 753.69 $\pm$ 44.83 | 5.34 $\pm$ 0.19 | 6.58 $\pm$ 0.13 | 82.73 $\pm$ 1.39 |
> |  Hard   | Ours           | 290.36 $\pm$ 69.56 | 124.61 $\pm$ 26.76 | 1326.60 $\pm$ 189.43 | 11.93 $\pm$ 1.31 | 12.36 $\pm$ 1.20 | 135.05 $\pm$ 8.18 |
> |     | MaskedMimic（Oracle）    | 47.74 $\pm$ 1.38   | 27.25 $\pm$ 0.80   | 829.02 $\pm$ 7.70    | 8.33 $\pm$ 0.09  | 10.60 $\pm$ 0.21 | 146.90 $\pm$ 6.64 |
> |     | **Ours（Oracle）** |        79.25 $\pm$ 34.70 | 34.74 $\pm$  11.30 | 734.90$\pm$  77.95 | 7.04 $\pm$  0.71 | 8.34 $\pm$  0.57 | 93.79 $\pm$  8.68 |
>
>
> We thank the reviewer again for the detailed, valuable feedback. We hope our responses have addressed all concerns, and we remain happy to answer any further questions.

---

> > ### Comment · Reviewer_339U · 2025-08-05
> >
> > I thank the authors for the detailed response.
> >
> > ## W2, W3
> > Thanks for the details. As agreed by the authors, there exist many alternative methods aiming to deal with motion feasibility filtering. Some use adaptive filtering with policy tracking response as critics, which I think is better justified compared with empirically selecting motions based on handcrafted criteria. The contribution of this component can be further reinforced by comparing it with the aforementioned baselines.
> >
> > ## W4
> > CMDP is a well-established formulation that has simple and standard algorithms. The simplest approximation using the Lagrangian multiplier is to learn a separate critic addressing the constraints. I would recommend that the authors look into this alternative to achieve the performance from a technically sound method.
> >
> > ## W5
> > The primary concern lies in the reasonableness of approximating the optimal tracking error with the current EMA estimate, which can be a strong assumption. This approximation is crucial as it determines the central claim of the design of the optimal tracking factor. It would be helpful to provide further information on how valid such an approximation is, with evaluations of expert policy performance.
> >
> > ## W6, W7
> > Thanks for the additional information and comparison. I do think the proposed method can shine with its motion filtering and adaptive learning strategy generalized to multi-motion setups, as single-motion tracking has been well studied in the field.

---

> > > ### Author Response · Authors · 2025-08-07
> > > **Response**
> > >
> > > We sincerely thank the reviewer for the thoughtful and constructive feedback.
> > >
> > >
> > > ## W2/W3: Motion Filter
> > >
> > > Following your advice, we conduct experiments comparing our rule-based motion filtering approach with the policy tracking-based filtering methods.
> > >
> > > We train two H2O teacher policies with privileged information, where (i) one policy is trained on the AMASS dataset and evaluated on our target filtering dataset, (ii) and the other one is trained and evaluated directly on our target filtering dataset. Consistent with previous experimental settings, we report the F1 score, precision, and recall for classifying motions as trackable (Episode Length Ratio > 60%) on the same dataset. The results are presented in the table below.
> > > |                              | Precision | Recall | F1 socre |
> > > |------------------------------|:---------:|:------:|:--------:|
> > > | H2O Teacher (AMASS)          |    0.66   |  1.00  |   0.80   |
> > > | H2O Teacher (Target Dataset) |    0.66   |  1.00  |   0.80   |
> > > | Ours                         |    1.00   |  1.00  |   1.00   |
> > >
> > > It can be observed that the tracking-based filtering method has relatively low precision, meaning some learnable motions are misclassified as unlearnable. We attribute this to multi-motion policies struggling with highly dynamic motions that single-motion policies can master, due to the purpose of generalizability. Consequently, this filtering method tends to incorrectly filter out such motions, making it unsuitable for our goal of tracking highly dynamic motions.
> > >
> > > ## W4: CMDP
> > >
> > > CMDP aims to learn a policy $\pi$ that maximizes the value function under certain constraints (e.g., a safety function $g$), formulated as:
> > > $\max_{\pi} V^{\pi}(s) \quad \text{s.t.} \quad g(s) < c$, which can be solved by converting it into an unconstrained problem and estimating the gradient by sampling trajectories according to the policy and dynamics. In contrast, our bi-level problem does not aim to solve for the constrained policy. Instead, it focuses on obtaining the optimal tracking factor $\sigma$ in closed form. As a result, it avoids the need to consider the discount factor of rewards and the stochastic nature of policies and dynamics.
> > >
> > > In practice, we consider the tracking errors $\mathbf{x}$ at the trajectory level, which eliminates the need to calculate cumulative rewards with a discount factor or sample experiences from stochastic dynamics. Such a formulation is desirable because we require a fixed tracking factor $\sigma^\*$ for the entire trajectory, which is different from the optimal policy $\pi^\*$ that changes at each step. Therefore, the lower-level optimization in our problem that solves for $\pi^\*$ is conducted at the trajectory level based on absolute policy performance. Moreover, we can solve the bi-level problem with a closed-form solution without gradient estimation, as is required in CMDP.
> > >
> > > We appreciate the reviewer's suggestion to use CMDP. However, using CMDP introduces additional complexities in problem formalization and gradient estimation. We emphasize that the tracking factor can be optimized in a trajectory level, avoiding the reliance on CMDP and makes the formulation an efficient solution.
> > >
> > >
> > > ## W5: EMA Approximation
> > >
> > > To validate the EMA as an approximation of expert performance, we evaluate the converged policy at 50k training steps, and compare its actual evaluation results with the EMA values. The table below reports tracking errors of key terms across representative motions, each setting with 3 seeds. The close alignment between EMA and evaluated values supports the reasonableness of this approximation.
> > >
> > >
> > > | Motion | Setting | Joint Position (1e-2) | Max Joint Position (1e-1) | Lower Body Position (1e-4) | Upper Body Position (1e-4) |
> > > |--------|--------|---------------------|-------------------------|--------------------------|--------------------------|
> > > | Fight | Evaluated | 1.61 $\pm$ 0.14 | 2.52 $\pm$ 0.09 | 5.58 $\pm$ 1.14 | 4.87 $\pm$ 0.99 |
> > > |  | EMA | 1.59 $\pm$ 0.04 | 2.48 $\pm$ 0.02 | 4.87 $\pm$ 0.10 | 5.46 $\pm$ 0.15 |
> > > | JumpKick | Evaluated | 2.55 $\pm$ 0.03 | 3.55 $\pm$ 0.11 | 12.56 $\pm$ 0.55 | 7.63 $\pm$ 0.23 |
> > > |  | EMA | 2.95 $\pm$ 0.04 | 3.77 $\pm$ 0.06 | 13.52 $\pm$ 0.57 | 12.56 $\pm$ 0.64 |
> > > | LPunch | Evaluated | 2.42 $\pm$ 0.14 | 3.84 $\pm$ 0.02 | 3.75 $\pm$ 0.24 | 3.96 $\pm$ 0.60 |
> > > |  | EMA | 2.64 $\pm$ 0.06 | 3.64 $\pm$ 0.06 | 4.63 $\pm$ 0.08 | 5.05 $\pm$ 0.30 |
> > > | Rspin | Evaluated | 2.08 $\pm$ 0.13 | 3.06 $\pm$ 0.09 | 10.79 $\pm$ 1.56 | 12.32 $\pm$ 1.75 |
> > > |  | EMA | 2.65 $\pm$ 0.02 | 3.40 $\pm$ 0.01 | 10.19 $\pm$ 0.37 | 12.45 $\pm$ 0.58 |
> > >
> > > ## W6/W7 Multi-motions
> > > We're glad the potential of our method in multi-motion settings is recognized. While we focus on single-motion training in this work, we agree that extending to multi-motion scenarios is a valuable direction and plan to explore it further in future work.
> > >
> > >
> > >
> > > We hope that our detailed responses and additional results address the reviewer’s concerns satisfactorily.

---

> > > > ### Comment · Reviewer_339U · 2025-08-07
> > > >
> > > > I thank the authors for the detailed explanation and the additional experiments. I think including these results in the paper will greatly reinforce its conclusion.

---

### Official Review · Reviewer_qDky · 2025-07-02

**Clarity:** 4
**Significance:** 3
**Originality:** 3
**Rating:** 5
**Confidence:** 3

**Summary:**

The paper studies the task of humanoid robotic imitation of full-body human motions. The authors first extract feasible motions from videos by combining prior approaches: they estimates candidate motions via GVHMR, then filter these motions based on the maximum distance between center of mass and center of pressure, apply a vertical correction, and then retarget motions to respect the joint limits of the G1 robot.

Next, they train policies in simulation (including domain randomization) to track these motions via RL. To do this, they use an exponential tracking reward, whose distance parameter sigma is adaptively reduced over time, providing coarse shaping at the beginning and more precise shaping as training progresses. The trained policies are evaluated in simulation against the prior methods OmniH2O and ExBody2, demonstrating substantial improvement.

Finally, policies are deployed on a real robot, demonstrating similar tracking performance as in simulation.

**Questions:**

1) Table 1 reports mean +/- one standard deviation. Presumably the goal is to identify which methods are significantly better than others; why not compute standard errors, and use them to report 95% CIs? (Or, if the data may not be normally distributed, you could report asymmetric CIs based on a nonparametric test, e.g. a bootstrap test.)

Bolding is currently done based on standard deviation of the best result, but this will underestimate errors; differences should be checked for significance using a test for difference in means, e.g. a t-test (or for non-normal data, a permutation test: https://docs.scipy.org/doc/scipy/reference/generated/scipy.stats.permutation_test.html).

2) Could you provide some additional intuition for the derivation of optimal tracking sigma? I understand why shrinking sigma is useful in training (to provide successively finer reward shaping), but for the theoretical derivation it sounds like you care about the converged policy (153); why isn’t sigma for the optimal policy arbitrarily large, so that it’s as close to linear as possible, thus best mimicking the true linear tracking error J^ex?

**Ethical Concerns:**

["NO or VERY MINOR ethics concerns only"]

**Final Justification:**

In the rebuttal discussion, the authors added CIs and statistical significance calculations, resolving my concerns here. They also added ablations as suggested by reviewer oUB5, and provided additional explanation for their theoretical results.

The limitation of separate policies for each motion remains unresolved, but this is a limitation acknowledged by the authors, and I believe the paper remains strong despite this.

I've read the review by reviewer 339U (who gave the lowest score), but the author's responses seem to address the concrete concerns, and I believe the paper's value rests on its strong empirical results for their simple and elegant method.

**Limitations:**

Yes

**Quality:**

3

**Strengths And Weaknesses:**

**Strengths:**
1) The adaptive motion tracking optimization is an elegant, theoretically-motivated solution to the reward shaping problem, a common problem in robotics. As shown in Figure 7, it avoids the need for task-specific tuning of tracking factor.
2) Evaluations demonstrate clear improvements over prior recent work (Table 1).
3) The full pipeline leads to impressive, highly dynamic motions achieved on the real robot, as shown in the supplementary videos.
4) The paper is clearly written overall.

**Weaknesses:**
1) Separate policies are trained for each motion, limiting generalization (acknowledged by authors in limitations).
2) Some of the statistical analysis seems nonstandard; see questions.

---

> ### Author Rebuttal · Authors · 2025-07-29
>
> We thank Reviewer qDky for the detailed and insightful feedback. We appreciate your recognition of the theoretical contribution, real-world results, and clarity of presentation. Below, we respond to your questions and concerns:
>
>
> # W1: Limited to single motion
>
> We thank the reviewer for highlighting this important limitation. While our current framework trains separate policies per motion, it can be naturally extended to multi-motion settings by incorporating reference motion states as inputs and modestly expanding the network architecture. We view this as a valuable direction for future work.
>
> # Q1: Statistical significance
>
> We have revised the Table 1 to report mean $\pm$ standard error (instead of standard deviation)  and include statistical significance testing between our methods and each baseline method. Specifically, we conducted two-sided permutation tests (as recommended) to assess whether the observed differences in means are statistically significant.
>
> Significant improvements (p < 0.05) by *Ours* over baselines are marked with asterisks (\*) in the table for clarity. The additional results of *Ours(Oracle)*, suggested by other reviewers, are also reported to compare with *MaskedMimic(Oracle)*.
>
> Across many metrics and difficulty levels, *Ours* shows statistically significant improvements over prior methods, confirming the robustness of our approach.
>
> | Level    | Method         | $E_{g-mpbpe}↓$      | $E_{mpbpe}↓$        | $E_{mpjpe}↓$          | $E_{mpbve}↓$       | $E_{mpbae}↓$      | $E_{mpjve}↓$        |
> | --- | -------------- | ------------------ | ------------------ | -------------------- | ----------------- | ---------------- | ------------------ |
> | Easy    | OmniH2O        | 233.54 $\pm$ 2.00* | 103.67 $\pm$ 0.95* | 1805.10 $\pm$ 6.15*  | 8.54 $\pm$ 0.06*  | 8.46 $\pm$ 0.04* | 224.70 $\pm$ 1.02  |
> |     | Exbody2        | 588.22 $\pm$ 5.71* | 332.50 $\pm$ 1.79* | 4014.40 $\pm$ 10.75* | 14.29 $\pm$ 0.08* | 9.80 $\pm$ 0.07* | 206.01 $\pm$ 0.67* |
> |     | Ours           | 53.25 $\pm$ 8.80   | 28.16 $\pm$ 3.06   | 725.62 $\pm$ 8.10    | 4.41 $\pm$ 0.16   | 4.65 $\pm$ 0.07  | 81.28 $\pm$ 1.02   |
> |     | MaskedMimic（Oracle）    | 41.79 $\pm$ 0.86   | 21.86 $\pm$ 1.02   | 739.96 $\pm$ 9.98*    | 5.20 $\pm$ 0.12   | 7.40 $\pm$ 0.17*  | 132.01 $\pm$ 4.47*  |
> |     | Ours（Oracle） |   45.02 $\pm$ 3.38 | 22.95 $\pm$ 7.61 | 710.30 $\pm$ 8.33 | 4.63 $\pm$ 0.79 | 4.89 $\pm$ 0.48 | 73.44$\pm$ 6.21 |
> | Medium    | OmniH2O        | 433.64 $\pm$ 7.25*  | 151.42 $\pm$ 3.28* | 2333.90 $\pm$ 22.14* | 10.85 $\pm$ 0.13  | 10.54 $\pm$ 0.07 | 204.36 $\pm$ 2.00  |
> |     | Exbody2        | 619.84 $\pm$ 11.70* | 261.01 $\pm$ 0.71* | 3738.70 $\pm$ 12.04* | 14.48 $\pm$ 0.07* | 11.25 $\pm$ 0.08 | 204.33 $\pm$ 0.97* |
> |     | Ours           | 126.48 $\pm$ 12.08  | 48.87 $\pm$ 3.38   | 1043.30 $\pm$ 46.67  | 6.62 $\pm$ 0.18   | 7.19 $\pm$ 0.11  | 105.30 $\pm$ 2.66  |
> |     | MaskedMimic（Oracle）    | 150.92 $\pm$ 59.65 * | 61.69 $\pm$ 20.58 * | 934.25 $\pm$ 69.35 *  | 8.16 $\pm$ 0.88*   | 10.01 $\pm$ 0.39* | 176.84 $\pm$ 11.68* |
> |     | Ours（Oracle） |       66.85 $\pm$ 22.49 | 29.56 $\pm$ 6.50 | 753.69 $\pm$ 44.83 | 5.34 $\pm$ 0.19 | 6.58 $\pm$ 0.13 | 82.73 $\pm$ 1.39 |
> |   Hard  | OmniH2O        | 446.17 $\pm$ 6.42  | 147.88 $\pm$ 2.07  | 1939.50 $\pm$ 11.94  | 14.98 $\pm$ 0.32 | 14.40 $\pm$ 0.29 | 190.13 $\pm$ 4.10  |
> |     | Exbody2        | 689.68 $\pm$ 5.90  | 246.40 $\pm$ 0.62* | 4037.40 $\pm$ 8.37*  | 19.90 $\pm$ 0.10 | 16.72 $\pm$ 0.08 | 254.76 $\pm$ 1.70* |
> |     | Ours           | 290.36 $\pm$ 69.56 | 124.61 $\pm$ 26.76 | 1326.60 $\pm$ 189.43 | 11.93 $\pm$ 1.31 | 12.36 $\pm$ 1.20 | 135.05 $\pm$ 8.18  |
> |     | MaskedMimic（Oracle）    | 47.74 $\pm$ 1.38   | 27.25 $\pm$ 0.80   | 829.02 $\pm$ 7.70*    | 8.33 $\pm$ 0.09 * | 10.60 $\pm$ 0.21* | 146.90 $\pm$ 6.64*  |
> |     | Ours（Oracle） |        79.25 $\pm$ 34.70 | 34.74 $\pm$  11.30 | 734.90$\pm$  77.95 | 7.04 $\pm$  0.71 | 8.34 $\pm$  0.57 | 93.79 $\pm$  8.68 |
>
>
>
>
> # Q2: Intuition of theoretical derivation
>
> > Why you are shrinking sigma in training time but talking about the converged policy for theoretical derivation?
>
> In short, our strategy of shrinking sigma during training is for approximately solving the theoretical formulation.
>
> In our theoretical derivation, we do care about the converged policy, and consider the problem as a bi-level optimization, where the lower level corresponding to RL training and the upper level corresponding to reward tuning. Meanwhile, in the implementation, we integrate these two levels into a single-stage RL training process with a practical adaptive mechanism (Sec.3.2.3).
>
> This integration is for practical considerations. While accessing to the precise performance of a specific sigma requires a converged policy, and iteratively train multiple policies to explore the suitable sigma is expensive, we use the EMA of tracking error during training to roughly access the converged performance. This provides a tractable and gradually accurate signal for adjusting sigma. As training proceeds and the policy converges, this approximation becomes increasingly reliable. Hence the converged policy trained with adaptive module should reflect the theoretical optimal solution.
>
> > Why isn’t sigma for the optimal policy arbitrarily large?
>
> While a larger sigma makes the $J^{in}$ mimicking the linear objective $J^{ex}$, this does not imply that the converged policy from the inner RL optimization will minimize $J^{ex}$. The bi-level formulation explicitly models this distinction.
>
> From a theoretical perspective, the lower level problem is subject to complex factors, including environment dynamics and implicit regularization, modelled by $R(x)$. These factors will bias the solution and prevent the learned policy from exactly minimizing $J^{ex}$. The upper level problem chooses sigma to improve the actual tracking outcome—not to match the functional form of the reward.
>
> From a practical perspective, setting sigma too large makes the reward too flat, weakening the learning signal. This makes the RL agent focus more on other rewards and leads to training failure. This is aligned with the above theoretical analysis.
>
>
>
> Thank you again for the thoughtful feedback and for recognizing the contributions of this work. If there are any questions you would like to discuss, please let us know.

---

> > ### Comment · Reviewer_qDky · 2025-08-01
> >
> > Thanks for your responses.
> >
> > **Q1: Statistical significance**
> >
> > Thanks, this addresses my concern. Agree that many of the improvements over prior work appear statistically significant.
> >
> > **Q2: Intuition of theoretical derivation**
> >
> > > From a practical perspective, setting sigma too large makes the reward too flat, weakening the learning signal. This makes the RL agent focus more on other rewards and leads to training failure. This is aligned with the above theoretical analysis.
> >
> > Right, this makes sense.
> >
> > > From a theoretical perspective, the lower level problem is subject to complex factors, including environment dynamics and implicit regularization, modelled by $R(x)$. These factors will bias the solution and prevent the learned policy from exactly minimizing $J^{ex}$. The upper level problem chooses sigma to improve the actual tracking outcome—not to match the functional form of the reward.
> >
> > Sorry, I’m still struggling a bit to understand the derivation in appendix A. A couple questions:
> >
> > 1) If there were no such theoretical environment dynamics (i.e. $R(x)=0$), then do you agree that the way to achieve an optimal theoretically converged policy with respect to the tracking outcome should be for the reward to best match the tracking outcome, hence arbitrarily large sigma?
> >
> > 2) If so, how does equation 17 depend on R? In particular, do we have the expected limiting behavior, where $\sigma \to \inf$ as $R(x) \to 0$?

---

> > > ### Author Response · Authors · 2025-08-03
> > > **Response**
> > >
> > > Thanks for your response.
> > >
> > >
> > > > 1.  If there were no such theoretical environment dynamics (i.e. $R(x)=0$  ), then do you agree that the way to achieve an optimal theoretically converged policy with respect to the tracking outcome should be for the reward to best match the tracking outcome, hence arbitrarily large sigma?
> > >
> > > If $R(x)=0$, the lower-level solution directly becomes $x^*=0$, so the value of sigma doesn't matter.
> > >
> > >
> > > > 2.  If so, how does equation 17 depend on R? In particular, do we have the expected limiting behavior, where $\sigma\rightarrow\inf$ as $R(x)\rightarrow 0$?
> > >
> > > We don't think R and sigma have such a simple relationship. The gradient behaviour of $R$, rather than its magnitude, plays a more important role.
> > >
> > > Intuitively speaking, since $x^\*$ is obtained from setting the derivative of lower level objective to zero (as Equation 11), sigma affects the solution by adjusting the balance of gradients, i.e., reducing the gradient at $x^\*$ to shift the solution.

---

> > > > ### Comment · Reviewer_qDky · 2025-08-04
> > > >
> > > > Thanks for your response. My questions have been addressed.

---

### Official Review · Reviewer_TKpF · 2025-07-03

**Clarity:** 3
**Significance:** 3
**Originality:** 3
**Rating:** 6
**Confidence:** 4

**Summary:**

This paper proposes PBHC, a physics-based motion imitation framework for humanoid robots to imitate highly dynamic human motions such as Kungfu and dancing. The method includes a motion processing stage to extract, filter, correct, and retarget motions, and an adaptive imitation stage inspired by a bi-level optimization and using an asymmetric actor-critic architecture. The framework is validated through simulation and real-world experiments, including a series of visually compelling demonstrations on the Unitree G1 robot.

**Questions:**

Providing the training time and learning curves for some representative motions could be helpful to the readers.

**Ethical Concerns:**

["NO or VERY MINOR ethics concerns only"]

**Final Justification:**

The authors have addressed my concerns. I change the rating to 6: strong accept.

**Limitations:**

Yes.

**Quality:**

3

**Strengths And Weaknesses:**

The paper provides strong real-world validation, demonstrating stable and expressive behaviors of the Unitree G1 robot in complex motion tasks. The idea of adaptively adjusting the tracking weight sigma is intuitive and seems very helpful.

Many implementation details are provided in the supplementary which is helpful to the community to learn sim-to-real. In addition, comprehensive ablation studies are performed.

Dynamically adjusting sigma is intuitive, but I have some concerns about the authors’ theoretical justification and claims.

Simply put, achieving a higher reward through reward design does not necessarily indicate a better policy.

Reward maximization should be the objective of the policy learning process itself. If we try to maximize the reward by modifying the reward function design, we can always change certain configurations to increase the reward value, but that does not imply improved reinforcement learning performance.
The optimization proposed in this paper aims to find an appropriate sigma under various conditions to increase the absolute value of the reward. However, this reward maximization does not guarantee a better policy, because the agent learns from temporal-difference (TD) errors, not just from receiving higher scalar reward values.
Specifically, if one action is better than another, a good reward function should provide clear distinctions to facilitate learning, rather than simply assigning a large reward to an action.

The reliance on contact masks limits the range of tasks that can be performed. It is primarily suited for foot-ground locomotion. Other agile motions such as climbing or rolling may become difficult or infeasible under current pipeline.

---

> ### Author Rebuttal · Authors · 2025-07-29
>
> We sincerely thank the reviewer for their thoughtful and constructive feedback. We are encouraged that the reviewer found our real-world validation strong, the dynamic tracking weight adjustment intuitive, and appreciated our detailed implementation and ablation studies. We address your concerns point by point below::
>
> # W1: Reward maximization != good policy
>
> We agree with your observation: simply increasing scalar reward values via reward shaping does not necessarily lead to a better policy, as policy learning is driven by TD errors and relative action advantages.
>
> However, we would like to clarify a possible misunderstanding of our formulation. Our method does not attempt to achieve a higher scalar reward by simply adjusting sigma. Rather, in our theoretical formulation (Sec.3.2.2), we model both the RL training and reward shaping into a bi-level optimization problem where:
> - The lower-level optimization, expressed as a simplified motion tracking problem, corresponds to the standard RL process, aiming to train a policy that maximizes the *fixed* tracking reward and other reward terms, given a specific sigma.
> - The upper-level optimization, which is not part of the RL loop, seeks the optimal sigma that minimizes the accumulated tracking error of the final converged policy. This outer optimization is *not reward maximization* but a performance-driven objective based on *absolute* external metrics.
>
> In short, sigma is chosen not to maximize reward per se, but to guide the RL process such that the resulting policy achieves better *external performance*.
>
> In the implementation, we integrate these two levels into a single-stage RL training process with a practical adaptive mechanism (Sec.3.2.3).
>
>
> # W2: Reliance on contact mask
>
> We agree that our current framework is best suited for motions involving foot-ground contact, and this is a limitation for tasks such as climbing or rolling. However, the contact mask formulation is inherently generalizable. In principle, the same technique can be extended to detect and handle contacts for other body parts (e.g., hands, pelvis, head). We believe this simple yet general technique holds promise for enabling the imitation of a broader range of contact-rich behaviors.
>
>
> # Q1: Learning curve and training time
>
> Thank you for the suggestion. We have provided training curves and wall-clock training times for representative motions in the following tables.
>
> *Training Time*: Aggregated across all main experiments since it's motion-agnostic.
>
> |  Iterations | 10K       | 20K       | 30K       | 40K| 50K       |
> |----------------------|-----------|-----------|-----------|----------|-----------|
> | Training Time (hours)| 5.38$\pm$ 0.03 | 10.70$\pm$ 0.11 | 15.97$\pm$ 0.22|21.25 $\pm$ 0.30|26.28 $\pm$ 0.54|
>
> *Learning curve of typical motions*: Each row summarizes mean $\pm$ standard error over 3 seeds.
> |  Motion  |  Metric  | 5K       | 10K       | 30K       | 50K       |
> |----------------------|----------------------|-----------|-----------|-----------|----------|
> |Jabs punch| Mean episode length| 193.33 $\pm$ 1.04 |  229.53 $\pm$ 0.81 |233.20 $\pm$ 0.40|234.97 $\pm$ 0.42|
> | | Mean reward| 8.02 $\pm$ 0.07 | 13.81 $\pm$ 0.49 |  14.41 $\pm$ 0.08|15.84 $\pm$ 0.38|
> | Tai Chi| Mean episode length| 169.67 $\pm$ 0.37 | 253.90 $\pm$ 1.76 | 358.83 $\pm$ 4.07|375.03 $\pm$ 1.33|
> | | Mean reward| 6.93 $\pm$ 0.41 | 9.89 $\pm$ 0.58 | 19.46 $\pm$ 1.35|22.19 $\pm$ 0.58|
> | Roundhouse kick| Mean episode length| 96.62 $\pm$ 7.75 | 120.50 $\pm$ 1.66 | 125.73 $\pm$ 0.07|127.37 $\pm$ 0.32|
> | | Mean reward| 3.82 $\pm$ 0.07 | 4.41 $\pm$ 0.26 | 6.26 $\pm$ 0.05|6.53 $\pm$ 0.08|
>
> We hope this addresses your concerns. We will revise the paper to clarify our formulation, and include the requested training analysis. Let us know if there is any other clarification we can provide.

---

> ### Comment · Reviewer_TKpF · 2025-08-05
>
> Thank you! My concerns have been addressed.

---

### Official Review · Reviewer_oUB5 · 2025-07-03

**Clarity:** 3
**Significance:** 2
**Originality:** 2
**Rating:** 4
**Confidence:** 5

**Summary:**

The paper proposes "filter - correct - retarget - reweight " pipeline for realizing robust humanoid imitation of highly dynamic human motions like kungfu and dancing. The core innovation here is to (1) Define a stability metric inspired from zero moment point to filter our unstable motions. (2) Correct the motion to always have at least one foot contact with the actual ground and shift the motion accordingly. (3) Dynamically Reweight individual weights of the exponential kernel in the decomposed reward function to facilitate a natural curriculum for tracking individual joints more aggressively as the imitation behavior improves. They prove the efficacy of the approach by showcasing experimental results with sim2real transfer for a wide range of highly dynamic skills on G1 platform. Results show that the their method with adaptive nature of reward function and reward vectorization together allow them to outperform strong baslines like OmniH20 and Exbody2.

**Questions:**

*Questions*
- I would love to know some thoughts from authors on the sim2real gap on G1. do you think the DR range suggested by the paper is enough and there is no sim2real gap ? What would you make out of papers like ASAP[1] which specifically deals with the sim2real gap of the ankle.


[1] He, Tairan et al. “ASAP: Aligning Simulation and Real-World Physics for Learning Agile Humanoid Whole-Body Skills.” ArXiv abs/2502.01143 (2025): n. pag.

**Ethical Concerns:**

["NO or VERY MINOR ethics concerns only"]

**Final Justification:**

I am inclined to accept as training dynamic motions for humanoids is overall a brewing research topic and the paper showcases evidence that their training methodology provides a meaningful improvement on being able to train policies for a non-perfect diverse and dynamic dataset.

**Limitations:**

Yes

**Quality:**

3

**Strengths And Weaknesses:**

*Strengths*
- The filter-correct-retarget-reweight paradigm is highly practical and well motivated. The authors clearly showcase that the dynamics based filtering is an efficient proxy to filter out "unrealizable" motions.
- The reweighting of individual reward components being paired with reward vectorization is very intuitive and well motivated. Ablation studies experimentally show that the adaptive reward tracking helps in overall performance especially in highly dynamic motions.
- The final output of the controller in sim and especially real world on highly dynamic motions are very impressive.
- The implementation details and other hyper-parameters are very detailed suggesting high chances of reproducibility.


*Weakness*
- No ablation study on reward vectorization. How crucial is the vectorization of rewards to make the adaptive reward mechanism to show benefits.
- It would be interesting to see how masked-mimic performs if the regularizations were added and privileged information removed ? Or we can remove the regularizations from the proposed method as well - as an ablation study. Just shrugging masked-mimic off as a oracle-style lower bound while being a good anchor does not provide us much information of how much performance do we lose to each of the regularization components and privileged informations.
- It would be interesting to further look into why the "unrealizable" motions are so. Did it happen due to pose tracking error from the video / Mocap data? In this context what kind of human motions even when perfectly tracked might be flagged as "unrealizable" ? Some discussion on this is very welcome.

---

> ### Author Rebuttal · Authors · 2025-07-29
>
> We sincerely thank the reviewer for their thoughtful and constructive feedback. We are glad that the reviewer found our proposed pipeline practical, well-motivated, and effective in enabling robust humanoid imitation of highly dynamic motions. We appreciate the positive comments on our real-world results, reward adaptation mechanism, and detailed implementation. We address the concerns raised point-by-point below.
>
> # W1: No ablation study on reward vectorization
>
> We conducted ablation studies comparing performance with and without reward vectorization on representative motions, each with 3 seeds. The results across different training steps are summarized below. Overall, reward vectorization facilitates training and consistently improves early learning performance, especially noticeable in the first 10K steps.
>
>
>
> | Motion           | Setting  | Metric              | 5K                | 10K               | 30K               | 50K               |
> |------------------|----------|---------------------|-------------------|-------------------|-------------------|-------------------|
> | Jabs punch       | w/ RV   | Mean episode length      | 193.33 $\pm$ 1.04      | 229.53 $\pm$ 0.81      | 233.20 $\pm$ 0.40      | 234.97 $\pm$ 0.42      |
> |                  |          | Mean reward         | 8.02 $\pm$ 0.07        | 13.81 $\pm$ 0.49       | 14.41 $\pm$ 0.08       | 15.84 $\pm$ 0.38       |
> |                  | w/o RV    | Mean episode length      |  173.93 $\pm$ 9.06             |  228.87 $\pm$ 0.85             | 232.43 $\pm$ 0.49             | 234.07 $\pm$ 0.37            |
> |                  |          | Mean reward         | 7.05 $\pm$ 0.51             |  13.06 $\pm$ 0.50             | 14.26 $\pm$ 0.10            | 15.75 $\pm$ 0.37             |
> | Tai Chi          | w/ RV   | Mean episode length      | 169.67 $\pm$ 0.37      | 253.90 $\pm$ 1.76      | 358.83 $\pm$ 4.07      | 375.03 $\pm$ 1.33      |
> |                  |          | Mean reward         | 6.93 $\pm$ 0.41        | 9.89 $\pm$ 0.58        | 19.46 $\pm$ 1.35       | 22.19 $\pm$ 0.58       |
> |                  | w/o RV    | Mean episode length      | 167.27 $\pm$ 1.42             | 254.43 $\pm$ 3.21             | 354.43 $\pm$ 0.77             | 372.87 $\pm$ 1.41             |
> |                  |          | Mean reward         |6.59 $\pm$ 0.44            | 10.29 $\pm$ 0.37             | 19.01 $\pm$ 0.53            | 22.07 $\pm$ 0.30             |
> | Roundhouse kick  | w/ RV   | Mean episode length      | 96.62 $\pm$ 7.75       | 120.50 $\pm$ 1.66      | 125.73 $\pm$ 0.07      | 127.37 $\pm$ 0.32      |
> |                  |          | Mean reward         | 3.82 $\pm$ 0.07        | 4.41 $\pm$ 0.26        | 6.26 $\pm$ 0.05        | 6.53 $\pm$ 0.08        |
> |                  | w/o RV    | Mean episode length      | 97.34 $\pm$ 2.06             |118.63 $\pm$ 0.41             | 124.83 $\pm$ 0.41            | 125.63 $\pm$ 0.27             |
> |                  |          | Mean reward         | 3.85 $\pm$ 0.48            | 4.40 $\pm$ 0.25             | 6.15 $\pm$ 0.05             | 6.46 $\pm$ 0.36            |
>
>
> # W2: Loss from regularization and privilege information
>
> As suggested, we conducted additional ablation experiments to enable a fairer comparison with MaskedMimic’s oracle setting. Specifically, we trained our method with privileged information and without domain randomization, which we refer to as *Ours (Oracle)*. The results are presented in the table below.
>
> It demonstrates that our method achieves competitive or superior performance to MaskedMimic under identical oracle settings.
>
>
> |  Level   | Method         | $E_{g-mpbpe}↓$    | $E_{mpbpe}↓$      | $E_{mpjpe}↓$       | $E_{mpbve}↓$     | $E_{mpbae}↓$     | $E_{mpjve}↓$       |
> | --- | -------------- | ---------------- | ---------------- | ----------------- | --------------- | --------------- | ----------------- |
> |  Easy   | Ours           | 53.25 $\pm$ 8.80 | 28.16 $\pm$ 3.06 | 725.62 $\pm$ 8.10 | 4.41 $\pm$ 0.16 | 4.65 $\pm$ 0.07 | 81.28 $\pm$ 1.02  |
> |     | MaskedMimic （Oracle）   | 41.79 $\pm$ 0.86 | 21.86 $\pm$ 1.02 | 739.96 $\pm$ 9.98 | 5.20 $\pm$ 0.12 | 7.40 $\pm$ 0.17 | 132.01 $\pm$ 4.47 |
> |     | **Ours（Oracle）** |   45.02 $\pm$ 3.38 | 22.95 $\pm$ 7.61 | 710.30 $\pm$ 8.33 | 4.63 $\pm$ 0.79 | 4.89 $\pm$ 0.48 | 73.44$\pm$ 6.21 |
> |  Medium   | Ours           | 126.48 $\pm$ 12.08 | 48.87 $\pm$ 3.38  | 1043.30 $\pm$ 46.67 | 6.62 $\pm$ 0.18 | 7.19 $\pm$ 0.11  | 105.30 $\pm$ 2.66  |
> |     | MaskedMimic (Oracle）    | 150.92 $\pm$ 59.65 | 61.69 $\pm$ 20.58 | 934.25 $\pm$ 69.35  | 8.16 $\pm$ 0.88 | 10.01 $\pm$ 0.39 | 176.84 $\pm$ 11.68 |
> |     | **Ours（Oracle）** |       66.85 $\pm$ 22.49 | 29.56 $\pm$ 6.50 | 753.69 $\pm$ 44.83 | 5.34 $\pm$ 0.19 | 6.58 $\pm$ 0.13 | 82.73 $\pm$ 1.39 |
> |  Hard   | Ours           | 290.36 $\pm$ 69.56 | 124.61 $\pm$ 26.76 | 1326.60 $\pm$ 189.43 | 11.93 $\pm$ 1.31 | 12.36 $\pm$ 1.20 | 135.05 $\pm$ 8.18 |
> |     | MaskedMimic（Oracle）    | 47.74 $\pm$ 1.38   | 27.25 $\pm$ 0.80   | 829.02 $\pm$ 7.70    | 8.33 $\pm$ 0.09  | 10.60 $\pm$ 0.21 | 146.90 $\pm$ 6.64 |
> |     | **Ours（Oracle）** |        79.25 $\pm$ 34.70 | 34.74 $\pm$  11.30 | 734.90$\pm$  77.95 | 7.04 $\pm$  0.71 | 8.34 $\pm$  0.57 | 93.79 $\pm$  8.68 |
>
>
>
>
> # W3: Unrealizable motion
>
> We identify two main sources of “unrealizable” motions:
> 1. Tracking artifacts: Motions from video or mocap may contain artifacts (e.g., teleporting, ground penetration, floating feet) due to reconstruction errors, which violate physical constraints and hinder learning. Imperfect motion retargeting can further exacerbate these issues.
> 2. Robot limitations: Even with perfect tracking, some motions are infeasible due to mechanical capacity limitations, such as high-speed action, excessive torque, or unstable contact transitions. (e.g., toe balancing in ballet or rapid one-leg spins).
>
>
> # Q1: Sim2real gap
> In our experiments, while our DR strategy effectively cover a wide range of scenarios, sim2real gap still exists, particularly for motions that heavily rely on ankle and waist pitch/roll actuation. We suspect this is partly due to the parallel-joint mechanical design of these modules, which introduces structural discrepancies between simulation and hardware. We also observe that robot-specific factors, including manufacturing variation and wear, influence sim2real performance.
>
> We view the ASAP paper as a valuable and complementary approach. Its delta action model effectively mitigates sim2real discrepancies—especially at the ankle—and could potentially improve success rates and tracking accuracy. However, our results show that even without such corrections, our method achieves strong sim2real performance across dynamic skills, suggesting that ASAP addresses an important but not exclusive aspect of the sim2real challenge.
>
>
>
> We hope the additional explanations helped address your concern. We will include the suggested analysis and comparison in the revised manuscript. We are happy to answer further questions that you may have.

---

> > ### Comment · Reviewer_oUB5 · 2025-08-04
> >
> > Thank you for the rebuttal and addressing all my concerns. I maintain my acceptance score as the authors have satisfactorily responded to the key issues raised.
> >
> > Regarding the reward vectorization ablation (W1), It is somewhat disappointing that the benefits are primarily limited to early training stages (first 10K steps). I would recommend including the wall-clock training times for these iterations in the main paper and discussing this temporal limitation more explicitly. Happy to see that the oracle comparison (W2) demonstrates competitive performance with MaskedMimic.

---

### Author Response · Authors · 2025-08-05
**Request for AC Assistance before Discussion Deadline**

Dear AC,

We sincerely appreciate your dedication to the conference and express our gratitude to all the reviewers for their valuable feedback. We have carefully considered all suggestions and updated our manuscript accordingly.

However, we have not yet received responses from Reviewers 339U and TKpF. With only three days remaining for discussion, we kindly request your assistance in reaching out to these reviewers. It would be greatly appreciated if you could encourage them to review our rebuttal, as we are eager to know if we have adequately addressed their questions and concerns.

We believe that constructive and timely communication between reviewers and authors is essential and beneficial for all parties involved.

Thank you for your hard work and support.

Best regards,

The authors

---

### Decision · Program_Chairs · 2025-09-17

**Decision:**

Accept (poster)

**Comment:**

The paper proposes an approach for humanoid robots to imitate dynamic full-body human movement demonstrations. The proposed approach consists of a pipeline with a smart combination of existing techniques together with following the humanoid optimized motions using domain randomized reinforcement learning. For adapting to the desired motions gradually, the paper uses a reward schedule that forces the humanoid motions closer to the desired trajectory as the training progresses.

Strengths of the paper include a well thought out framework combining existing techniques with additional advances resulting in high performance humanoid motions, strong performance compared to baselines, real world robot experiments, and a well written paper. One potential weakness is that policies are not general but limited to each motion.

Overall, the paper provides a scientific advance in humanoid motion imitation and experimental evidence including demonstrations with a real robot.

During the rebuttal the authors provided stronger statistical support for the results, further comparisons and ablation studies. The rebuttal phase resulted in resolving most concerns.